# Crowd-sourced trait data can be used to delimit global biomes

Simon Scheiter[1], Sophie Wolf[2], and Teja Kattenborn[3]

[1]Senckenberg Biodiversity and Climate Research Centre (SBiK-F), Senckenberganlage 25, 60325 Frankfurt am Main, Germany
[2]Remote Sensing Centre for Earth System Research (RSC4Earth), Leipzig University and Helmholtz Centre for Environmental Research, Talstr. 35, 04103 Leipzig, Germany
[3]Department for Sensor-based Geoinformatics, Faculty of Environment and Natural Resources, University of Freiburg, Tennenbacherstr. 4, 79106 Freiburg, Germany

**Correspondence:** Simon Scheiter (simon.scheiter@senckenberg.de)

**Abstract.** Terrestrial biomes and their biogeographic patterns have been derived from a large variety of variables including species distributions, bioclimate or remote sensing products. Yet, classifying the biosphere into biomes from a functional perspective using biophysical traits has rarely been tested. Such a trait-based biome classification was limited by data availability. Here, we aimed to exploit crowd-sourced plant observations and trait databases to assess systematically which traits are most suitable for biome classification. We derived global patterns of 33 biophysical traits covering around 50% of the land surface by combining crowd-sourced species distribution data from GBIF and trait observations from the TRY database. Using these trait maps as predictors for supervised cluster analyses, we tested to what extent we can reconstruct 31 published biome maps. A sensitivity analysis with randomly sampled combinations of traits was performed to identify traits that are most appropriate for biome classification. Performance was quantified by comparing modeled biome maps and the respective observation-based biome maps. Finally, spatial gaps in the resulting biome maps were filled using species distribution models to obtain continuous global biome maps. We showed that traits can be used for biome classification and that the most appropriate traits are conduit density, rooting depth, height, and different leaf traits, including specific leaf area and leaf nitrogen content. The best performance of the biome classification was obtained for biome maps based on biogeographic zonation and species distributions, in contrast to biome maps derived from optical reflectance. The availability of crowd-sourced plant observations is heterogeneous and, despite its exponential growth, large data gaps are prevalent. Nonetheless, it was possible to derive biome classification schemes from these data to predict global biome patterns with good agreement. Therefore, our analysis is a valuable approach towards understanding biome patterns based on biophysical traits and associated ecological strategies.

## 1 Introduction

Biomes are commonly used to represent major vegetation formations based on functional and structural attributes, and to map their biogeographic distributions (Moncrieff et al., 2016; Higgins et al., 2016). They have been widely used to study past, present, and future vegetation change and attribute the effects of respective drivers such as climate change or land use (e.g., Allen et al., 2020; Martens et al., 2020; Scheiter and Savadogo, 2016). Due to the high level of aggregation of vegetation features, biomes are useful to illustrate vegetation change in the context of climate mitigation, conservation and management.

Multiple biome maps were developed based on a variety of different data sources (Beierkuhnlein and Fischer, 2021). These include biogeographic zonation based on species distributions, bioclimatic and edaphic variables, or a variety of remote sensing products such as vegetation indices, optical reflectance or vegetation height. Depending on the variables used for biome classification, biomes reflect specific ecological functions associated with those variables (Mucina, 2019). Recently, Fischer et al. (2022) aggregated 31 different biome and land cover maps, showing considerable differences between classification methods, the biome types represented in the maps and their spatial distributions. In addition to species distribution data or remote sensing data, plant traits provide a detailed representation of vegetation at the site level. Plant traits are attributes of plants, describing for example their morphology, phenology or physiology, that determine the functioning of plants in their environment (Kattge et al., 2020). Traits are coordinated along trade-off axes, defining the spectrum of plant form and function (Díaz et al., 2016), and ecological strategies characterized by combinations of different traits vary between biomes and climatic zones (Pierce et al., 2017). It has been argued, that traits should be included in biome classification (Mucina, 2019; Hunter et al., 2021), and previous studies showed that trait data are suitable for delineating plant functional types (PFTs, Verheijen et al., 2016) and biomes (van Bodegom et al., 2014; Boonman et al., 2022; Scheiter et al., 2024). Despite the increasing availability of trait data in databases such TRY (Kattge et al., 2020) and extrapolated global trait maps (Wolf et al., 2022; Boonman et al., 2020), a systematic assessment of the performance of traits for biome classification and an identification of the most appropriate traits remain elusive. A functional trait perspective can provide novel biome maps with a more plant-oriented characterization of biomes and provide insights into the functional strategies and ecophysiology characterizing biomes.

In a recent study, Boonman et al. (2022) used height, specific leaf area (SLA), and wood density to reproduce the Olson et al. (2001) biome map by applying a supervised cluster analysis. Global maps for these traits were empirically derived by extrapolating trait data from the TRY database using statistical and machine-learning approaches (Boonman et al., 2020). This modeling approach allowed to reproduce current biome patterns and project future biome patterns under various climate change scenarios based on traits. Scheiter et al. (2024) used traits simulated by a dynamic vegetation model, the aDGVM2 (Langan et al., 2017; Kumar et al., 2021), and 31 different biome maps provided by Fischer et al. (2022, hereafter F31 maps) for biome classification. Similarly to Boonman et al. (2022), biome clustering was conducted by applying supervised cluster analyses, and a trait ranking revealed that traits related to size were most important for biome classification. While both studies showed that traits can be used for biome classification, both approaches show caveats.

Boonman et al. (2022) used only a set of three different traits - height, wood density and SLA - even though data on more traits are available in TRY (Kattge et al., 2020). The selection of traits was mainly driven by methodological criteria rather than by ecological knowledge, and ensured that traits could be extrapolated to the global scale with sufficient predictive accuracy using their statistical method. Extrapolation of geographically sparse TRY data to the global scale is uncertain (Ludwig et al., 2023), and Dechant et al. (2023) showed significant discrepancies between global trait maps created using different extrapolation methods. Such uncertainties in empirical trait maps may propagate to uncertainties in biome classification.

Trait maps simulated by the aDGVM2 are provided as continuous maps for the simulated study region (Scheiter et al., 2024). Limitations of using model results are model uncertainties in the aDGVM2, mismatches between observed and simulated trait patterns and, accordingly, uncertainties in biome classification. Further, some of the traits simulated by the aDGVM2 are

difficult to observe in reality (e.g., the amount of carbon allocated to different plant compartments), while other traits are not

represented in the model (e.g., conduit density or seed length). Therefore, classification schemes derived from aDGVM2 cannot be directly applied to real-world situations. These limitations may bias the suitability of different traits modeled in Scheiter et al. (2024) for biome classification.

A further caveat of both studies is that trait values of all species in plant communities were averaged, while different PFTs or species groups were not considered separately in the classification. Given that trait expressions or trait distributions of grasses

and trees, for example, differ substantially, we expect that such averaging reduces the performance of biome classification compared to a classification where traits of different PFTs are considered separately. In summary, the caveats of previous studies suggest that trait-based biome clustering should be conducted with a large number of observation-based traits, ideally interpolated to global spatial scale with high accuracy.

Recently, Wolf et al. (2022) presented a novel approach to create global maps for a series of traits of vascular plants by merg-

ing crowd-sourced species distribution data from iNaturalist (inaturalist.org) with trait data from the TRY database (Kattge et al., 2020). iNaturalist is a citizen science project for plant species identification that builds on smartphone apps. To enhance the spatial extent of trait data, this approach was now extended by integrating species distribution data from the Global Biodiversity Information Facility (GBIF). Moreover, trait patterns were derived separately for non-woody species (grasses and herbs), woody species (trees and shrubs), and for all species combined. We used these trait maps and supervised cluster anal-

yses to create trait-based biome maps informed by each of the F31 observation-based biome maps compiled by Fischer et al. (2022). We asked: (1) are trait patterns derived by combining heterogeneous crowd-sourced species distribution data and the TRY trait data appropriate to delimit global biomes despite substantial data gaps? (2) Which and how many traits are appropriate and required to delimit global biomes? (3) Does considering traits of different species groups improve biome classification? (4) Which trait values are characteristic for different biomes across all F31 biome maps? The trait data were not available for

the entire global land surface but covered only around 50% of the land surface. Accordingly, the biome maps from the cluster analyses had the same spatial gaps. To obtain continuous global trait-based biome maps, we applied species distribution models for extrapolation.

## 2 Materials and Methods

### 2.1 Data

We used trait patterns of vascular plants derived from combining GBIF species observation data and trait data from the TRY database (Kattge et al., 2020, vers. 6, Fig. 1). GBIF is a free and open access database containing biodiversity data such as species occurrence records. The TRY database contains $> 15$ million trait records for more than 300,000 plant taxa. The methodology follows Wolf et al. (2022), but instead of using iNaturalist data, a repository of crowd-sourced species distribution data, we used the entire GBIF archive to increase geographic and taxonomic coverage. Species observations ($n = 257,357,303$)

were downloaded from GBIF on June 2nd, 2023 (GBIF.Org User, 2023). This download already included filtering for 'Observation', 'Human observation' or 'Occurrence' records with no geospatial issues, no related records, a minimum distance

of 1,500 m to a country centroid and the occurrence status set to 'present', since GBIF contains true absence data. Additional filtering using the package R 'CoordinateCleaner' (Zizka et al., 2019) removed records with a coordinate uncertainty > 10km and a precision > 0.1°, those located in the ocean and those that match common issues, such as records that are falsely given coordinates along the equator or central meridian, and made sure to only include records identified to the species level. The observations were then linked to the TRY gap-filled dataset via species names (Schrodt et al., 2015), which resulted in a total of $n = 192,667,225$ observations. 90% of the filtered GBIF observations and 24% of species in GBIF were matched with 70% of species in TRY (numbers based on map products using all plant functional types). We spatially sub-sampled the data to limit both computational load and data redundancy: the matched observations were binned into equal area hexagons of 2,591 km$^2$ using the package 'dggridR' (Barnes and Sahr, 2023), which corresponds to about 0.5° at the equator. From each hexagon, we then sampled 10,000 observations. If a hexagon contained less than 10,000 observations, all observations were kept. After this sub-sampling 31,808,221 observations remained. These trait observations were then aggregated using a mean function to a global raster grid with 0.5°spatial resolution. We created three different maps for each trait in this manner, each for different combinations of plant species: 1) grasses and herbs (hereafter non-woody plants), 2) trees and shrubs (hereafter woody plants) and 3) all species (i.e., non-woody and woody plants). Trait distributions of non-woody and woody plant species differ substantially, for example in height. Calculating trait means for different species groups therefore allows the consideration of understorey versus overstorey and community composition in our analysis. Therefore, we filtered the observations according to plant functional types (PFTs) before spatially aggregating the trait values, where the association of a plant species to non-woody or woody plant PFTs was based on a majority vote of the PFT information contained in the TRY database (trait ID 197). Filtering traits before aggregation and not afterward, keeps as much trait information from the TRY data in the mean trait values as possible. All traits included in the analysis are provided in supplementary Table S1. The trait data created by this method do not cover the entire land surface, and the spatial coverage is shown in Fig. S1. Excluding Antarctica, coverage of the trait data was 49.6% (32,238 grid cells with trait data, 65,023 in total at 0.5° spatial resolution).

Fischer et al. (2022) compiled 31 different biome and land cover maps (Fig. 1). We aggregated the maps from the 10×10 km spatial resolution to the 0.5° resolution of the trait data using the 'raster' R package (Hijmans, 2020). As the biome maps are categorical, we used the nearest neighbor method for aggregation. We removed biomes that occur in less than 40 grid cells and grid cells not covered by vegetation such as water and built-up areas.

## 2.2 Biome clustering

Following an approach previously applied by Boonman et al. (2022) and Scheiter et al. (2024), we used a supervised cluster analysis to assign a biome type to each grid cell using trait information in the grid cell. Specifically, we used Gaussian mixture models ('MclustDA' function in 'mclust' R package, Scrucca et al., 2016) and fitted models for each of the F31 biome maps separately (Fig. 1). Clustering was only conducted for subsets of the 33 available traits (up to 12 traits, see next sections for selection of subsets). Analyses including all 33 trait simultaneously were not conducted due to the low performance and failure of the clustering algorithm. As described above, the trait data had large gaps and covered only around 50% of the land surface. Accordingly, the biome maps created by the cluster analysis only covered the spatial extent of the trait data (Figs. S1, 1).

The clustering of biomes was tested with different species-specific trait maps (see section 2.1) considering 1) only non-woody species, 2) only woody species, and 3) all species (i.e., non-woody and woody species). We expect that clustering using all species (case 3) shows the best performance because it considers the entire plant community in a grid cell. In contrast, we expect that using only traits of non-woody plants shows the lowest performance, because these traits do not reflect wood-dominated communities such as forests adequately. In addition, we 4) combined traits of non-woody and woody plants (i.e., combining traits of non-woody plants in case 1 and traits of woody plants in case 2). In case 4, the number of traits in the cluster analysis was twice the number of traits in cases 1, 2 and 3. We expect, that the performance of the clustering is maximized in case 4, because it considers the non-woody and woody plant communities separately by their respective traits, rather than averaging traits of all species in the plant community. In addition, doubling the number of variables increases the level of information in the cluster analysis and should therefore improve the performance.

To assess the performance of the biome clustering, we used the $\kappa$ statistics (Monserud and Leemans, 1992) to compare the biome maps derived from the cluster analysis and the respective F31 map used to inform the clustering. As we did not use the results from the clustering for spatial extrapolation, we used all available data for both the cluster analysis and the calculation of $\kappa$, without splitting the data into training and testing data sets. The $\kappa$ value quantifies the agreement between categorical data sets. It considers the likelihood that agreement can occur by chance and is more robust than calculating overlap. Values $< 0$ indicate no agreement, between 0 and 0.2 slight agreement, between 0.2 and 0.4 fair agreement, between 0.4 and 0.6 moderate agreement, between 0.6 and 0.8 substantial agreement and between 0.8 and 1 almost perfect agreement.

## 2.3 Trait ranking for classification

To test which of the 33 traits obtained by combining TRY and GBIF data (sec. 2.2) are most suitable for biome classification, we used a randomized approach with a variable number of traits and randomly sampled sets of traits (Fig. 1). Using all possible combinations of 33 traits for the cluster analysis was computationally not feasible. For each set of traits, cluster analyses were conducted for all of the F31 maps individually and for four combinations of species (see sec. 2.1). The number of traits in the clustering ranged between 3 and 12. The upper limit of 12 traits was selected as the performance of the clustering saturated when increasing the number of traits (see results). Hence, cluster analyses for non-woody, woody plants and all species (cases 1, 2, and 3 in sec. 2.1) included 3 to 12 variables, whereas analyses with non-woody and woody plant traits combined (case 4 in sec. 2.1) included 4, 6, ..., 24 variables. We repeated the clustering until at least 600 models were available for each F31 biome map (21,867 models in total, Table S2). For each cluster analysis, the $\kappa$ value was calculated to quantify the performance of the clustering and its response to the number of traits included.

To identify the traits that are most suitable for biome classification, we ranked the traits following an approach previously used by Scheiter et al. (2024). We selected all models with substantial agreement (i.e., $\kappa > 0.6$) from the 21,867 models of the randomized sensitivity analysis with separate trait values for non-woody and woody plants (case 4 in sec. 2.1). We used only models with separate trait values for non-woody and woody plants because we expected the highest performance of the cluster analyses. For the subset of models, we counted how often each trait was included and expressed this number as percent value

relative to the number of models with substantial agreement. We interpret this value as a measure of the suitability of traits for
biome classification and ranked traits according to this measure.

We used a similar approach to rank the traits separately for each number of traits considered in the clustering while aggregating models for all F31 maps. This analysis revealed, if the set of suitable traits is related to the number of traits in the clustering or if the same set of traits had high rank irrespective of the number of traits included. The $\kappa$ values did not exceed 0.6 for low numbers of traits (see results). We therefore calculated the 90% percentile of the $\kappa$ values for each number of traits, and used only models with $\kappa$ greater than the 90% percentile to derive the ranking.

## 2.4 Performance of specific trait subsets

We assessed how the performance of the clustering was related to the F31 biome maps used to inform the clustering, to the set of traits used for the clustering and to the method for aggregating non-woody and woody plants to calculate trait means. Therefore, the cluster analyses were repeated for selected subsets of traits and for each of the F31 biome maps separately. Using subsets of traits was necessary and reasonable, because using the full set of 33 traits for the cluster analysis was computationally not feasible. In addition, the sensitivity analysis showed, that the importance of traits differs and that performance of the cluster analyses saturates with an increasing number of traits included (see results). These findings suggest that a subset of traits is sufficient to obtain cluster analyses with high performance. A lower number of traits simplifies the interpretation of our results, e.g., on trait co-variation or specific trait combinations in biomes. Finally, the accuracy and spatial cover of trait observations differs, such that the selection of high-quality traits is reasonable. Therefore, we systematically selected three different subsets based on the trait ranking in sec. 2.3, hereafter denoted clusters. Cluster 1: as the trait ranking in sec. 2.3 was conducted for up to 12 traits, we selected the 12 traits with the highest rank in this analysis (see results section). This selection ensured substantial data-model agreement for some of the F31 biome maps and $\kappa > 0.6$, while constraining the number of traits included. Cluster 2: to constrain the number of traits, we selected 6 traits from the 12 traits in cluster 1. To account for traits describing different components of the plant economic spectrum (Díaz et al., 2016) and those commonly used in the literature, and to avoid inclusion of similar or redundant traits (e.g., several traits describing leaf nitrogen), we selected wood density, rooting depth, SLA, height, isotopic leaf nitrogen and conduit density (also see results section). Cluster 3: the number of traits was further constraint and included only 3 trait. Specifically, clustering was conducted for wood density, height and SLA as these traits are commonly included in trait-based studies and in a previous biome classification by Boonman et al. (2022). Using the same set of traits allows comparisons of different approaches. While our selection of traits was mainly based on the sensitivity analysis and trait ranking, the analysis can be conducted for any selection of traits relevant in specific case studies.

For a selected biome map and trait cluster, we created a confusion matrix and calculated biome-specific $\kappa$ values. The confusion matrix represents the overlap between all different combinations of modeled and observation-based biomes. For each biome, those count values were normalized by the total number of grid cells covered by a biome in the observation-based map. The confusion matrix identified biomes that are well represented or misclassified by the cluster analysis. For each biome, the $\kappa$ values were calculated individually. Here, we selected results from the cluster analysis with trait cluster 1 including 12 traits informed by the Nature Conservancy (2009) biome map, as this map had a high $\kappa$ value (see results).

## 2.5 Biome-specific traits and model performance

Biome types used in different F31 biome maps differ substantially between maps. Therefore, biome-specific mean trait values and $\kappa$ values cannot be compared directly across all biome maps. In our study, the F31 biome maps were not aggregated into a consensus biome map. Instead, we calculated biome-specific mean trait values by selecting all biomes across the F31 maps that share similarity in their names. Specifically, we merged all biome names of the F31 biome maps as provided in the supplementary materials of Fischer et al. (2022) in one text string. Then we counted the occurrences of all words in this string. After removing unnecessary words such as 'and' or 'with', we obtained a list of attributes defining different biomes such as 'forest', 'evergreen' or 'boreal'. For the most frequent attributes (Table S3), we selected all biomes from all F31 maps that contained this attribute in its name and calculated the mean trait values and $\kappa$ for each of those biomes. We first conducted the analyses for the three trait clusters with 12, 6, and 3 traits (sec. 2.4) and for models including non-woody and woody plant trait separately (case 4 in sec. 2.2). However, this analysis included up to 24 traits (12 traits for both non-woody and woody plants), and we simplified the analysis by using trait values of all species (case 3 in sec. 2.2). To calculate biome-specific $\kappa$ values, we transformed the observation-based and the modeled biome maps into a binary map where one represents the target biome and zero all other biomes. Then, we calculated $\kappa$ for these binary maps.

This analysis showed that 'forest', 'tropical' and 'temperate' were used most frequently in the biome names with 179, 99 and 84 occurrences. Attributes such as 'forest' included all forests from the boreal to the tropical zone such that we expected large variation of the traits. We repeated the counting procedure separately for biomes containing each of those three attributes. This procedure provided combinations such as 'evergreen forest' or 'tropical savanna'. For the most frequent combinations of attributes (Table S4), we calculated the biome-specific mean trait values and $\kappa$ values. Note, that this approach ignores biome maps that do not use any of these attributes to denote biomes such as the Higgins et al. (2016) map, or the Netzel and Stepinski (2016) map that denotes different biomes as numbered clusters. This analysis was conducted only for trait cluster 1, as it showed the highest data-model agreement when $\kappa$ values for all F31 biomes are averaged.

To assess associations between traits and biome attributes described in the previous paragraphs, we conducted principal component analyses (PCA). For the PCA, we used traits of the three trait clusters in sec. 2.4 (i.e., 12, 6, and 3 traits) and, to reduce the number of traits included in the analysis, mean trait values of all species (non-woody and woody plants, case 3 in sec. 2.2). For each biome attribute, we first identified all biomes in all of the 31 biome maps, that contained the attribute in the biome names. Then, for each of those maps, trait means for all grid cells covered by the target biome were calculated. By this procedure, we obtained for example 179, 53 and 37 trait means for the attributes 'forest', 'desert' and 'savanna', respectively, or 99 and 84 values for the attributes 'tropical' and 'temperate', respectively (Table S3). These trait means were then used for the PCA. We created PCAs for different sets of biome attributes defining for example the climate zone, biome type, or different forest types (Tables S3, S4). We used the 'ggbiplot' package (Vu and Friendly, 2024) to create biplots of the PCA results. To illustrate the location of biomes according to the biome attributes used calculate mean trait values, we color-coded points with the attributes in the biplots.

## 2.6 Continuous global biome maps from traits

The trait data derived from combining GBIF and TRY, and accordingly the biome maps obtained from the cluster analyses, did not cover the entire global land surface (Fig. S1). To obtain a global biome map with full coverage and to assess if heterogeneous and sparse trait data can be used to predict global biomes, we extrapolated the biome maps from the cluster analyses to the full global coverage by using species distribution models (SDMs, Franklin, 2009, Fig. 1) and bioclimatic variables (Booth et al., 2014) from WorldClim (Hijmans, 2020). Here, only the biome map obtained from 12 non-woody and woody plant traits (case 4, cluster 1) informed by the Nature Conservancy (2009) biome map was used to create a SDM. Yet, this extrapolation can be conducted for any of the F31 biome maps and trait clusters. For the SDMs, we selected a subset of the bioclimatic variables: mean annual temperature (bio1), mean annual precipitation (bio12) and further uncorrelated variables (correlation<0.6, variables bio2, bio7, bio10, bio14, bio15, bio18). For each biome of the considered biome map, an ensemble of 36 models was fitted using GLM, CTA, ANN, SRE, FDA and RF, three different sets of 3,500 randomly selected pseudo-absences and two replicates. Then, an ensemble was derived by including all models with True Skill Statistic TSS>0.5 and by combining models calculating the mean suitability value. By conducting predictions with the ensemble model and the 10-minute WorldClim data, we derived global suitability maps for the different biomes. Finally, suitability maps with continuous suitability values per biome were aggregated to a categorical biome map by identifying for each grid cell the biome with the highest suitability value.

The extrapolated global biome map with full coverage derived from the SDM was validated. Therefore, it was compared to the corresponding observation-based biome map used to create the SDM using the $\kappa$ statistics and TSS, and a confusion matrix was constructed to quantify the overlap between modeled and observation-based biomes (see sec. 2.4). Niche models were fitted using the 'biomod2' R package (Thuiller et al., 2023).

## 3 Results

### 3.1 Trait ranking for biome classification

The randomized sensitivity analysis showed that both the median and the maximum $\kappa$ value increased when the number of traits included in the cluster analysis increased (Fig. 2). The maximum $\kappa$ value was 0.64 for clustering with 11 traits for non-woody and woody plants (22 traits in total). The median $\kappa$ values saturated for higher numbers of traits, although the maximum value increased further. For a given number of traits, maps based on non-woody plant traits (case 1 in sec. 2.2) showed the lowest predictive performance. Higher model performance was obtained with trait maps derived from woody species, all species and non-woody and woody species combined (cases 2, 3, 4, respectively, Figs. 2, S2).

When selecting cluster analyses with substantial agreement ($\kappa > 0.6$), leaf carbon, isotopic leaf nitrogen and SLA had the highest rank, i.e., they were included in most models (in 41.1%, 40.3%, and 40.3% of the models, Fig. 3). Traits related to seeds, leaf area and leaf mass had the lowest rank and were included in less than 30% of the models, seed number in only 12.6%. Despite the considerable variation among the relevance of the traits, we did not observe that traits are overly suitable

or unsuitable for predicting biomes. The set of suitable and unsuitable traits for biome classification was robust for different numbers of traits included in the clustering and similar sets of traits had high or low rank irrespective of the number of traits (Fig. S3). Conduit density, isotopic leaf nitrogen, leaf nitrogen per area and SLA were among the highest-ranked traits for each number of traits in the sensitivity analysis, while seed number had the lowest rank for all cases.

## 3.2 Specific trait clusters

Data-model agreement varied considerably for different biome maps and trait clusters used for the clustering (Fig. S4). When averaged for all maps (averages of columns in Fig. S4), cluster analyses using traits of non-woody and woody plants separately showed higher $\kappa$ values than cluster analyses using only trait means of non-woody plants, only woody plants or all species. Using only traits of non-woody plants showed the lowest performance for all trait clusters. The highest average performance was obtained for cluster 1 (12 traits with highest rank in Fig. 3) with traits of non-woody and woody plants combined ($\kappa$=0.50). When averaging all trait clusters for each biome map (averages of rows in Fig. S4), the highest performance was achieved for the Nature Conservancy (2009) biome map ($\kappa$=0.48) and the lowest performance for the Tateishi et al. (2011, 2014) map ($\kappa$=0.21). We therefore selected the Nature Conservancy (2009) biome map for analyses focusing on a single biome map (sec. 2.4, 2.6).

When considering different trait clusters and biome maps individually, the $\kappa$ value was maximized when clustering was performed for the Zhang and Yan (2014) biome map with non-woody and woody plant traits of cluster 1 ($\kappa$=0.64), followed by the Nature Conservancy (2009) and the Olson et al. (2001) maps ($\kappa$=0.63). Observation-based biomes were best predicted in Europe, North America, parts of the Sahel region and Australia (Fig. 4). The biome types disagreed primarily in the subtropics. This result is also reflected in the biome-specific $\kappa$ values of the clustering across the F31 maps (Fig. 5, S5-S8), where $\kappa$ values were on average lower in tropical forests than in temperate or boreal forests. It is also reflected in the confusion matrix for the Nature Conservancy (2009) map, where $\kappa$ values and percent overlap were highest for Mediterranean forest, woodland and scrub, temperate broadleaf and mixed forest, and boreal forest/taiga (Table 1). $\kappa$ values and overlap were low for mangrove, (sub)tropical dry broadleaf forest and flooded grassland and savanna, and large proportions of these biomes were wrongly classified as (sub)tropical moist broadleaf forest.

For cluster 2 (wood density, rooting depth, SLA, height, isotopic leaf nitrogen and conduit density) and cluster 3 (SLA, wood density and height), performance was maximized when using traits of non-woody and woody plants separately and the Nature Conservancy (2009) biome map. The $\kappa$ values were 0.57 and 0.47, respectively.

## 3.3 Biome-specific traits

When selecting all biomes from the F31 maps that share similar attributes in their names, we found that mean trait values differ between biome types (Figs. S9-S13). For example, tundra vegetation was characterized by shallow roots, low wood density and high conduit density whereas savannas had deep roots and lower conduit density.

The attribute 'forest' was included in the names of 179 biomes in the F31 maps, and traits showed large variation across different forest types (Fig. S9). Splitting forests according to additional attributes revealed differences of mean trait values

between forest types (Fig. S11). For example, tropical and subtropical forest showed deepest roots whereas needleleaf forest and boreal forest showed shallow roots. Deep or shallow roots co-occurred with lower or higher SLA and conduit density, respectively. Similarly, splitting tropical and temperate biomes according to additional attributes revealed differences (Figs. S12, S13).

The PCA with 12 traits that had a high rank and with trait means calculated for biomes that share attributes describing the climatic zone, explained 47.5 and 27.2% of the variation on the first two axes (Fig. 6). For example, biomes in boreal areas showed higher conduit density but lower rooting depth, wood density and isotopic leaf N, in contrast to tropical regions. A PCA with trait means calculated for biomes that share attributes describing biome types (e.g., forest, savanna, grassland) explained 48.1 and 27.2% of the variation on the first two axes (Fig. S14). The first axis was mainly associated with conduit

density, isotopic leaf N and rooting depth, the second axis with leaf N, leaf CN and leaf thickness. The distribution of biomes in trait space overlapped substantially because attributes such as 'forest' included the entire range from boreal to tropical forests. Splitting forest by additional attributes showed a separation between boreal and temperate forests, but large overlap of, for example, the attribute 'evergreen forest', as it occurs in different climate zones (Figs. S15-S17). The patterns were similar when only 6 or 3 traits were used for the PCA (not shown).

**3.4   Predicting global biome patterns**

Using SDMs, global biome maps with full spatial coverage can be predicted based on biome patterns derived from cluster analyses using trait data (Fig. 7). The extrapolation step was evaluated by comparing modeled and underlying observation-based biome maps using the $\kappa$ statistics. Similarly to biome patterns modeled for the spatial extent of the trait data, data-model agreement between the global maps and the corresponding F31 map strongly differed. For example, for the Nature Conservancy

(2009) map, $\kappa = 0.70$ and for the Tateishi et al. (2011, 2014) map, $\kappa = 0.25$. The confusion matrix for the biome map based on the Nature Conservancy (2009) biome map revealed highest $\kappa$ values and overlap for tundra with 97.1% correctly modeled grid cells and lowest values for mangrove and flooded grassland and savanna (Table 2).

**4   Discussion**

We used trait maps derived from combining crowd-sourced species distribution data and plant trait data from the TRY database

to test if trait distribution maps can be used to delineate global biomes. Although the coverage of trait data was heterogeneous and sparse in the tropics and sub-tropics, and in the high northern latitudes of Canada and Siberia, the analyses showed that such a biome classification is promising. Using traits for biome classification represents a valuable approach as it based directly on biophysical and biochemical properties of the biomes, expressed by plant traits. Assessing biome distributions based on these traits provides better understanding of the ecological strategies that characterize biomes, as well as trait co-variation and

functional diversity within biomes. Yet, the agreement between modeled and observation-based biome maps strongly depends on the traits and the biome map used to develop biome classification schemes.

## 4.1 Suitability of traits for biome classification

The performance of biome classification using traits was strongly related to the number and the selection of traits used for the clustering. Previous studies on trait relationships (Wright et al., 2004; Díaz et al., 2016; Bruelheide et al., 2018) and biome classification using traits (van Bodegom et al., 2014; Boonman et al., 2022) focused on a limited number of traits and had a strong focus on leaf traits. Our results indicate that data-model agreement increases with the number of traits included in the analysis. This is not surprising given that additional traits add information to the clustering and represent additional trade-off axes in the trait space. A saturation of the performance for an increasing number of traits included indicates that the major trade-off axes between traits that characterize biomes can be captured with around 10 traits or more. Saturation occurs because the traits included in the analysis are correlated (Fig. S18) and hence contain redundant information: when removing a trait from the analyses, its information may still be represented by another, correlated trait. Yet, some traits and trait combinations were selected more often in best-performing cluster analyses and selection of a lower number of suitable traits can lead to high performance. In contrast, selecting unsuitable traits can lead to poor performance even if a high number of traits was used.

According to our analysis, height and leaf traits such as SLA, leaf carbon and leaf nitrogen were included in a high proportion of the models with the highest performance and can be considered as most suitable for biome classification. This finding confirms the existence of generic plant strategies (Grime, 1988; Pierce et al., 2017) and the spectrum of plant form and function, where size and leaf economic traits were identified as the two major axes of the trait space (Díaz et al., 2016). The results highlight the importance of wood and root traits, particularly conduit density and rooting depth. Higher conduit density has been associated with higher hydraulic conductivity, a higher risk of embolism (Martínez-Vilalta et al., 2012), and with gymnosperm species that invest more carbon into hydraulic safety (Yang et al., 2022). Root traits gained more attention recently (Chave et al., 2009). They are often linked to plant hydraulics and water availability (Anderegg et al., 2018), which makes them particularly important to understand how the projected increase of droughts under future climate change (IPCC, 2021) may influence ecosystem dynamics and resilience. Yet, measuring rooting traits is generally more difficult than measuring aboveground traits, which constrains the geographic coverage and taxonomic representativeness of root observations. According to our ranking, reproductive traits, particularly seed number, were least appropriate for biome classification. We attribute this result to the low coverage of seed traits in the TRY database, resulting in lower agreement between trait data derived by merging TRY and GBIF data with independent sPlot trait data (Wolf et al., 2022; Sabatini et al., 2021). Seed number may also be determined by the dispersal mechanism of plants, a trait not considered in our analysis. Dispersal mechanisms are influenced for example by small-scale heterogeneity of landscapes, microclimate or biotic interaction networks (Schleuning et al., 2016).

We conducted a similar sensitivity analysis with traits simulated by the dynamic vegetation model aDGVM2 (Scheiter et al., 2024). This previous study showed that traits related to plant size were included in most models with high performance, while leaf economic traits including SLA were less relevant. This contrasts the results of the study using observation-based trait data. Verheijen et al. (2016) used kernel density estimation to represent PFTs in trait space and found that four to five traits are sufficient to classify a large proportion of PFT correctly. A trait ranking revealed that specific rooting length was most

important, yet, it was not among the most suitable traits according to our study. Other traits, including height, leaf nitrogen or SLA had a high suitability rank in both studies, showing that similar traits are relevant for both PFT and biome classification.

## 4.2 Selection of biome map for clustering influences model performance

Using different trait clusters for biome classification revealed consistent differences of data-model agreement for different biome maps used to inform the clustering. When averaging all trait clusters considered in the cluster analyses, the Nature Conservancy (2009) map was best reproduced by the cluster analyses while the Tateishi et al. (2011, 2014) map showed the lowest performance. The maps assembled by Fischer et al. (2022) were derived from different data sources, including biogeographic zonation, species distributions, bioclimatic variables or remote sensing products including optical reflectance or NDVI. The performance of the clustering was high for biome maps based on biogeographic zonation and species distribution data (e.g., Nature Conservancy, 2009; Olson et al., 2001; Dinerstein et al., 2017) and lowest for biome maps based on optical reflectance (e.g., Tateishi et al., 2011, 2014). This can be explained by data sources in our study: we used species distribution data provided by the GBIF database and trait information from TRY. These sources resemble those of biome maps using species distribution data, in contrast to biome maps based on reflectance. Cluster analyses including only traits with a direct imprint on Earth observation signals, such as optical reflectance (Cherif et al., 2023; Kothari et al., 2023; Aguirre-Gutiérrez et al., 2021) may show higher agreement with such biome maps. Further, the remote sensing perspective from optical Earth observation satellites primarily informs on the top-canopy, while a representative biome characterization may require information on entire plant communities, including the understory (Dechant et al., 2023). Cluster analyses including 12 traits with the highest rank informed by the Zhang and Yan (2014) biome map showed high performance. This map was created by a cluster analysis of bioclimatic data and the high agreement underscores the relation between climate, traits and biomes (van Bodegom et al., 2014). Yet, the biome patterns in the Zhang and Yan (2014) map show substantial caveats. For example, the expansion of the central African tropical forest is heavily underestimated and other forest such as the Southern African Miombo are not represented. Therefore, 'tropical grassland' and 'Tropical Sahel and semidesert grassland' expand from the equator to the North and South (following biome names used in Fischer et al., 2022).

The maximum $\kappa$ value in the randomized sensitivity analysis was 0.64, indicating substantial agreement. Yet, $\kappa$ values higher than this maximum were not achieved. We attribute this upper limit to the heterogeneity of trait data available for different biomes, large differences in area covered by different biomes, and differences in the accuracy of different traits by merging TRY and GBIF data. When calculating $\kappa$ individually for different biomes, higher values were possible. For example, in the analysis for the Nature Conservancy (2009) map, $\kappa$ values for different biomes ranged between 0.25 and 0.79 in the cluster analysis, and some of the biomes with low data coverage (mangrove, flooded grassland and savanna) had low $\kappa$ values. The $\kappa$ value for the biome map with global coverage created by the SDMs increased to 0.7 when considering all biomes and was maximized for tundra ($\kappa$=0.84). The high performance in the tundra is surprising, given that data coverage was low for this biome type. This finding indicates that trait diversity in this biome is lower than in other biomes and captured by the trait data used in our study.

### 4.3 Relation of biome attributes and traits

According to our analysis, different biomes can be distinguished across the F31 maps based on trait values and expression biome-specific trait combinations. Our results show co-variation of different traits across biomes in the trait space. For example, tropical forests were characterized by the tallest vegetation, low SLA, high leaf nitrogen content and deep roots. These traits represent dominance of tall trees with evergreen phenology (typically associated with low SLA). Low SLA has been associated with conservative leaf strategies (Díaz et al., 2016). Tall vegetation indicates intense light competition in tropical forests. Deep roots can indicate drought avoidance (Oliveira et al., 2021) and rooting niche separation between tall trees with deep roots and understory vegetation with shallow roots (Walter, 1971; Ludwig et al., 2004). In the tropics and sub-tropics, substantial amounts of water may percolate into deeper soil layers and enhance rooting-zone water storage (Stocker et al., 2023). Deep roots ensure access to these water reservoirs. Deep roots and height were correlated and had the highest loadings on the same PCA axis because deep roots support mechanic stability of tall trees. Shallow roots in boreal forests indicate that those ecosystems are less limited by water or nutrients and more by light (Jonard et al., 2022). Yet, for other forest types, biome-specific trait means overlapped, such that the classification for these biomes is ambiguous. These results suggest, that the separation between different forest types is essential for an accurate classification of global biomes. To delineate biomes, we cannot rely on a few traits but need to consider the co-variation in a multivariate trait space and trait optimization towards multiple ecological functions.

Our approach to analyze biome-specific trait values by selecting biomes across all F31 biome maps that share similar attributes has caveats. Specifically, when selecting, for example, all biomes that share the attribute 'forest', then all forest from the boreal to the tropical zone are included. This explains the large variation of traits within this biome in our analysis, and the large overlap of different biomes in the PCA. Selecting other set of attributes such as the climatic zone or selecting biomes based on multiple attributes such as 'tropical forest' or 'boreal forest' allows a better separation. For example, it clearly showed higher conduit density in boreal biomes than in tropical biomes. Despite this caveat, our approach allows an integrated analysis of traits in multiple biome maps without aggregating all biome maps into a consensus map or reclassifying biomes into a lower number of mega-biomes (Champreux et al., 2024). Such approaches are often not objective because reclassification is not necessarily unique, and reclassification removes information from the biome maps.

### 4.4 Recommendations for trait-based biome classification

We showed that using trait data for biome classification is feasible, but that several decisions regarding data and methods are necessary. These decisions are related to the trait data, the traits included in the analysis, the biome map used to develop a classification, and the method used for clustering.

Multiple methods such as machine learning algorithms or niche models considering bioclimatic variables have been applied to extrapolate traits from the site level to the global scale. The resulting trait maps can differ substantially as each data set is affected by different biases and uncertainties (Dechant et al., 2023; Wolf et al., 2022; Schiller et al., 2021). Accordingly, biome maps derived from different trait maps can differ. We therefore advise selecting trait data with high accuracy for biome

classification and developing biome classification schemes for specific applications or regions using available and appropriate trait data.

In this study, trait information from TRY was extrapolated to larger areas by linking observed traits from TRY and observed species distributions from GBIF. This method was presented previously (Wolf et al., 2022; Schiller et al., 2021) and showed unprecedented agreement with independent observational data on community weighted traits (sPlot, Sabatini et al., 2021). The advantage of using these trait maps is that it includes gridded trait data for 33 different traits at much larger spatial scale than the original site data of these traits in TRY (possibly aggregated to 0.5° resolution), and all traits were available for the same spatial extent. Conducting our analyses with the original TRY site data, instead of gap-filled TRY data, would require sites or grid cells where all 33 traits are available. Further, TRY only represents trait observations obtained from single plants and these observations are known to be not representative of species distributions and plant communities, and have a large spatial bias (Kattge et al., 2020). The trait maps obtained from coupling TRY and GBIF represent mean trait values of entire plant communities (Wolf et al., 2022) and mirror the abundances of plant species in the communities.

Both the number of traits and the selection of traits influenced performance. Performance saturated as the number of traits increased. Yet, the selection of traits matters and including many unsuitable traits can imply low performance. Our results suggest that traits in biome classification should include leaf, stem and root traits as well as traits related to size to reflect different trade-off axes in the trait space. Here, we identified wood density, rooting depth, SLA, height, isotopic leaf nitrogen and conduit density as suitable traits. Ideally, the selected traits are only weakly correlated to avoid redundancy of information in the clustering. Yet, focusing only on weakly correlated traits can lead to poor performance if they do not reflect the global spectrum of plant form and function (result not shown).

When using traits for biome classification, data-model agreement was better when clustering was informed by biome maps based on vegetation data and biogeographic zonation than for maps based on remote sensing data. We therefore recommend using vegetation-based biome maps in the context of trait-based biome classification, such as Olson et al. (2001), Nature Conservancy (2009) or Dinerstein et al. (2017).

Here, we used Gaussian mixture models for the cluster analysis. Multiple alternative approaches are available, both for supervised and unsupervised classification. A systematic assessment of alternative approaches has not been performed, but may be valuable to identify the most appropriate method. Unsupervised classification can be used to create novel biome maps that are only defined by trait information but not by existing biome or land-cover maps. Alternatively, dimensionality reduction methods such as principal component analysis or isometric feature mapping could be applied to explore a trait-based, continuous representation of biome patterns.

## 5 Conclusions

The crowd-sourced trait data utilized for the analysis was spatially heterogeneous with large gaps in parts of the tropics, sub-tropics, and high northern latitudes. Nonetheless, the data were suitable for delimiting global biomes using cluster analyses and to predict global biome maps. This result highlights the value of crowd-sourced trait- and species distribution data in

the biogeography of biomes, despite the data gaps. We argue that filling gaps in the trait data would not only yield a more comprehensive understanding of the spectrum of plant form and function, but also allow more accurate biome classification. We also showed that increasing the number of traits in the cluster analysis improved model performance, highlighting the need to fill data gaps with respect to traits available for specific species or sites. While we assessed mean trait values of different biomes, this study focuses on methodological aspects. There is, however, large potential to further analyze the trait data generated by combining TRY and GBIF, for example to elucidate patterns of functional trait diversity and trait covariation within biomes. The trait data generated by combining TRY and GBIF data includes ranges of traits per grid cell and are suitable for such studies. Finally, our results can inform the development of dynamic vegetation models as it showed which traits are important for biome classification and which trait values are characteristic for different biomes. While leaf economic traits are already well-captured in such models, we argue that the representation of wood and root traits should be improved.

*Code and data availability.* Code and data that support the findings of this study are openly available.

1) Trait data: https://github.com/tejakattenborn/GBIF_trait_maps/

2) Bioclimatic data for species distribution models: https://www.worldclim.org

3) Fischer et al. (2022) biome maps: https://doi.org/10.5061/dryad.hqbzkh1jm.

4) All R scripts required to conduct the analyses and generate plots are available in Zenodo: https://zenodo.org/records/10526277. Access is currently restricted but will be granted to editors and referees on request. The repository will be made open access if manuscript is accepted for publication.

*Author contributions.* SW and TK created the trait data; SS conceived the study, performed the analyses and wrote the first draft of the manuscript; SW and TK contributed to the analyses and writing of the manuscript.

*Competing interests.* The authors declare that they have no conflict of interest.

*Acknowledgements.* TK acknowledges funding by the Deutsche Forschungsgemeinschaft (DFG, Emmy Noether project PANOPS, grant number 504978936). SW was funded by the National Research Data Infrastructure Germany for Biodiversity, NFDI4Biodiversity, a project by the German Research Foundation (DFG), project no. 442032008. The study was supported by the TRY initiative on plant traits (http://www.try-db.org). The TRY initiative and database are hosted, developed and maintained by J. Kattge and G. Boenisch (Max Planck Institute for Biogeochemistry, Jena, Germany), currently supported by Future Earth/bioDISCOVERY and the German Centre for Integrative Biodiversity Research Halle-Jena-Leipzig (iDiv, DFG-FZT 118, 202548816).

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

**Table 1.** Confusion matrix for biomes derived from the cluster analysis. Rows represent observation-based biomes, columns modeled biomes. Numbers represent the % of grid cells where data and model overlap, relative to the total number of the observed biome (i.e., counts were divided by the total number of grid cells covered by a biome in the observation-based biome map). $\kappa$ values were calculated individually for each biome. Here, results from the cluster analysis informed by the Nature Conservancy (2009) biome map were used as an example. Diagonal elements with correctly classified biomes are highlighted in bold font.

| Biome | $\kappa$ | 1 | 2 | 3 | 4 | 5 | 6 | 7 | 8 | 9 | 10 | 11 | 12 | 13 | 14 |
|---|---|---|---|---|---|---|---|---|---|---|---|---|---|---|---|
| 1 (sub)tropical moist broadleaf forest | 0.56 | **63.6** | 0.7 | 13.5 | 9.3 | 1.8 | 0.3 | 3.1 | 1.4 | 0.4 | 2.8 | 1.8 | 0.7 | 0.4 | 0.1 |
| 2 mangrove | 0.39 | 43.8 | **32.9** | 11 | 9.6 | 0 | 0 | 1.4 | 1.4 | 0 | 0 | 0 | 0 | 0 | 0 |
| 3 (sub)tropical grassland savanna and shrub | 0.64 | 15.2 | 0.1 | **69.1** | 3.4 | 0.2 | 0.7 | 5.9 | 1.9 | 0.1 | 1.2 | 1.8 | 0.3 | 0.1 | 0 |
| 4 (sub)tropical dry broadleaf forest | 0.35 | 30.1 | 0 | 12.9 | **39.5** | 5.1 | 0 | 7.5 | 2.3 | 0 | 0.8 | 1.3 | 0.5 | 0 | 0 |
| 5 (sub)tropical coniferous forest | 0.56 | 15.2 | 0 | 4.1 | 6.9 | **66.8** | 0 | 3.7 | 0 | 0 | 0.9 | 0.9 | 0.9 | 0.5 | 0 |
| 6 flooded grassland and savanna | 0.25 | 21.6 | 0 | 24 | 4.8 | 0.8 | **22.4** | 4.8 | 2.4 | 0.8 | 8 | 8.8 | 0 | 1.6 | 0 |
| 7 desert and xeric shrub | 0.63 | 6.3 | 0 | 9 | 5.8 | 1.3 | 0.2 | **64.2** | 1.7 | 2 | 1.2 | 6 | 1.6 | 0.7 | 0 |
| 8 montane grassland and shrub | 0.36 | 6.8 | 0 | 10.9 | 3.1 | 2 | 0.2 | 10.3 | **38.1** | 2 | 5.5 | 9.9 | 4.6 | 5.5 | 0.9 |
| 9 mediterranean forest woodland and scrub | 0.79 | 0.9 | 0 | 0.9 | 0.5 | 0 | 0 | 6.7 | 1.7 | **81.7** | 3.4 | 3.6 | 0.6 | 0 | 0 |
| 10 temperate broadleaf and mixed forest | 0.72 | 2.2 | 0 | 0.5 | 0.1 | 0.1 | 0.3 | 0.6 | 1.5 | 2.4 | **74.5** | 6.8 | 3.2 | 7.4 | 0.4 |
| 11 temperate grassland savanna and shrub | 0.63 | 3.1 | 0 | 1.9 | 0.8 | 0 | 0.4 | 7.3 | 2 | 0.9 | 8.6 | **68.0** | 2.4 | 3.9 | 0.5 |
| 12 temperate conifer forest | 0.58 | 1.7 | 0 | 0.3 | 0.2 | 0 | 0.3 | 6.9 | 1.4 | 1.9 | 11.1 | 5.9 | **56.0** | 12.2 | 2.2 |
| 13 boreal forest/taiga | 0.70 | 0.5 | 0 | 0 | 0 | 0 | 0 | 0.2 | 0.7 | 0 | 8.2 | 3.5 | 3.5 | **76.3** | 7.1 |
| 14 tundra | 0.67 | 0 | 0 | 0 | 0 | 0 | 0 | 0 | 1.2 | 0 | 1.6 | 1.1 | 0.5 | 23.3 | **72.4** |

**Table 2.** Confusion matrix for biomes derived from the species distribution model at full global coverage. Rows represent observation-based biomes, columns modeled biomes. Numbers represent the % of grid cells where data and model overlap, relative to the total number of the observed biome (i.e., counts were divided by the total number of grid cells covered by a biome in the observation-based biome map). $\kappa$ values were calculated individually for each biome. Here, results from the species distribution modeling informed by the Nature Conservancy (2009) biome map were used as an example. Diagonal elements with correctly classified biomes are highlighted in bold font.

| Biome | $\kappa$ | 1 | 2 | 3 | 4 | 5 | 6 | 7 | 8 | 9 | 10 | 11 | 12 | 13 | 14 |
|---|---|---|---|---|---|---|---|---|---|---|---|---|---|---|---|
| 1 (sub)tropical moist broadleaf forest | 0.71 | **76.5** | 8.2 | 3.7 | 3.7 | 1 | 2 | 0 | 2.1 | 0 | 2.8 | 0 | 0 | 0 | 0 |
| 2 mangrove | 0.09 | 50.4 | **35.3** | 11.2 | 1.3 | 0 | 0.6 | 1.1 | 0 | 0 | 0 | 0 | 0 | 0 | 0 |
| 3 (sub)tropical grassland savanna and shrub | 0.68 | 20.1 | 0.7 | **69.9** | 0.3 | 0.3 | 4 | 3.6 | 0.5 | 0 | 0.2 | 0.3 | 0 | 0 | 0 |
| 4 (sub)tropical dry broadleaf forest | 0.30 | 23.2 | 8 | 32 | **23.5** | 3.8 | 4.7 | 2.9 | 1.8 | 0 | 0 | 0 | 0 | 0 | 0 |
| 5 (sub)tropical coniferous forest | 0.36 | 31.3 | 3.1 | 6.8 | 1.1 | **49.9** | 1.3 | 1.5 | 3.5 | 0 | 1.1 | 0.4 | 0 | 0 | 0 |
| 6 flooded grassland and savanna | 0.05 | 15.8 | 0.2 | 38.8 | 0.1 | 0 | **9.3** | 12.5 | 0 | 0.4 | 1.6 | 21 | 0 | 0.2 | 0 |
| 7 desert and xeric shrub | 0.75 | 0.3 | 0 | 7.4 | 1 | 0.4 | 0.3 | **71.5** | 1 | 2.4 | 0.1 | 14.5 | 0.6 | 0.5 | 0.1 |
| 8 montane grassland and shrub | 0.57 | 1.1 | 0 | 2.5 | 0.1 | 3.9 | 0.1 | 6.7 | **53.5** | 1.8 | 0.6 | 6.4 | 2.4 | 9.4 | 11.5 |
| 9 mediterranean forest woodland and scrub | 0.65 | 0 | 0 | 0 | 0 | 0 | 0 | 10.5 | 0.9 | **83.1** | 4.6 | 0.2 | 0.7 | 0 | 0 |
| 10 temperate broadleaf and mixed forest | 0.71 | 2.1 | 0 | 0 | 0 | 0.1 | 3.6 | 1.7 | 0.7 | 4.1 | **68.6** | 9.1 | 2.2 | 6.9 | 0.8 |
| 11 temperate grassland savanna and shrub | 0.51 | 0.3 | 0 | 0 | 0.2 | 0 | 2.7 | 9 | 2.7 | 6.2 | 9.7 | **58.2** | 1.7 | 9 | 0.3 |
| 12 temperate conifer forest | 0.52 | 0.3 | 0 | 0 | 0.2 | 0.4 | 2.4 | 2.9 | 4.8 | 4.8 | 9.5 | 4.6 | **42.1** | 20.6 | 7.5 |
| 13 boreal forest/taiga | 0.74 | 0 | 0 | 0 | 0 | 0 | 0 | 0 | 0 | 0 | 1.4 | 0.3 | 0.2 | **74.4** | 23.8 |
| 14 tundra | 0.84 | 0 | 0 | 0 | 0 | 0 | 0 | 0 | 0 | 0 | 0 | 0 | 0 | 2.9 | **97.1** |

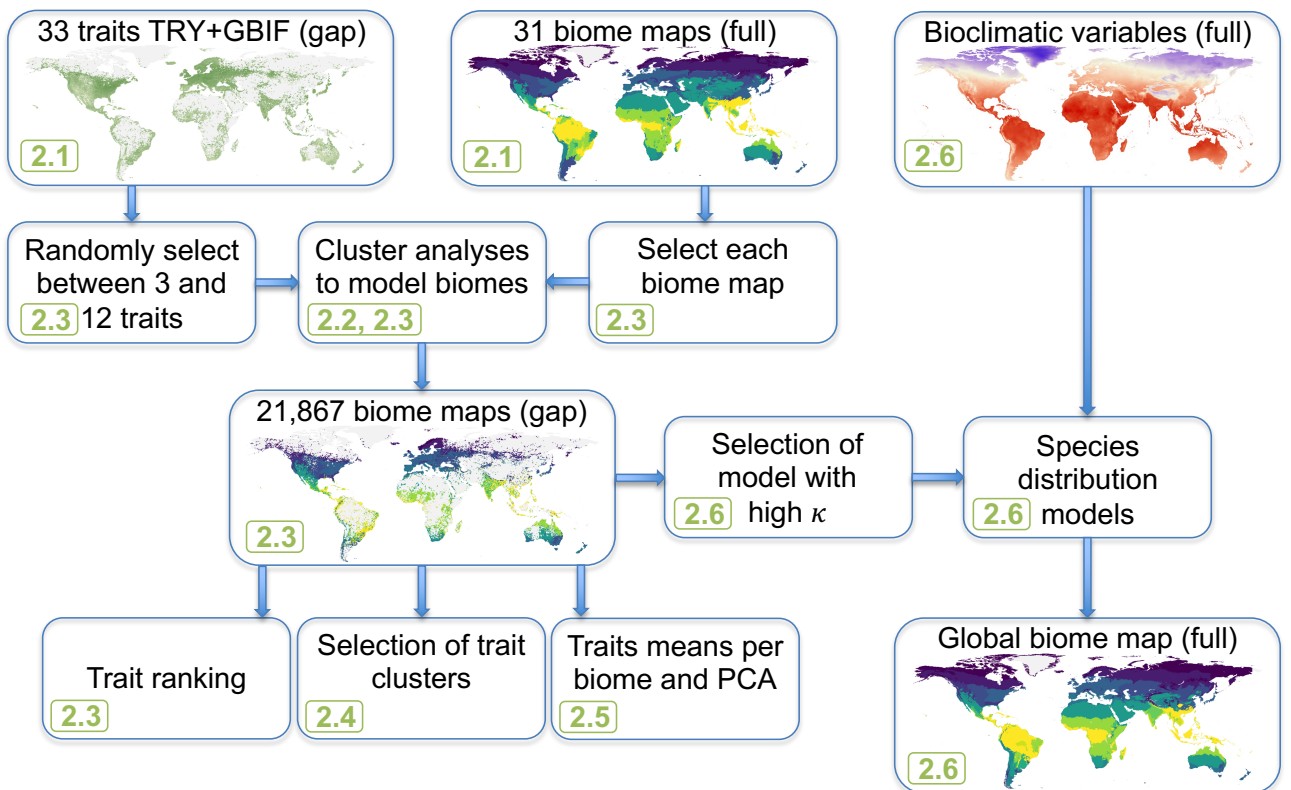

**Figure 1.** Modeling workflow of the study. Green numbers provide the section where the different steps are described, 'full' and 'gap' indicate if the data cover the entire land surface or only ca. 50% of the land surface where trait data are available.

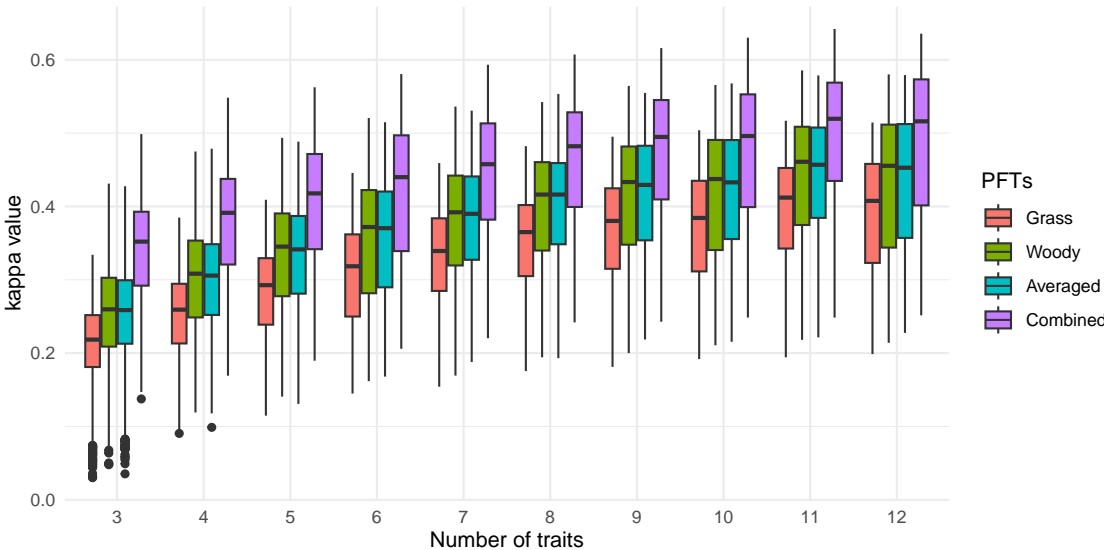

**Figure 2.** Relation between data-model agreement and the number of traits included in the cluster analysis. For each number of traits, the traits were randomly selected, and clustering was conducted for all F31 biome maps. At least 600 cluster analysis were conducted for each F31 map. Traits were sampled from those provided in Table S1 for different combinations of PFTs. Data-model agreement is represented by the $\kappa$ statistics. Note that 'Combined' indicates clustering with traits of both non-woody and woody plants, such that the number of traits is twice the number provided in the figure. See also Fig. S2.

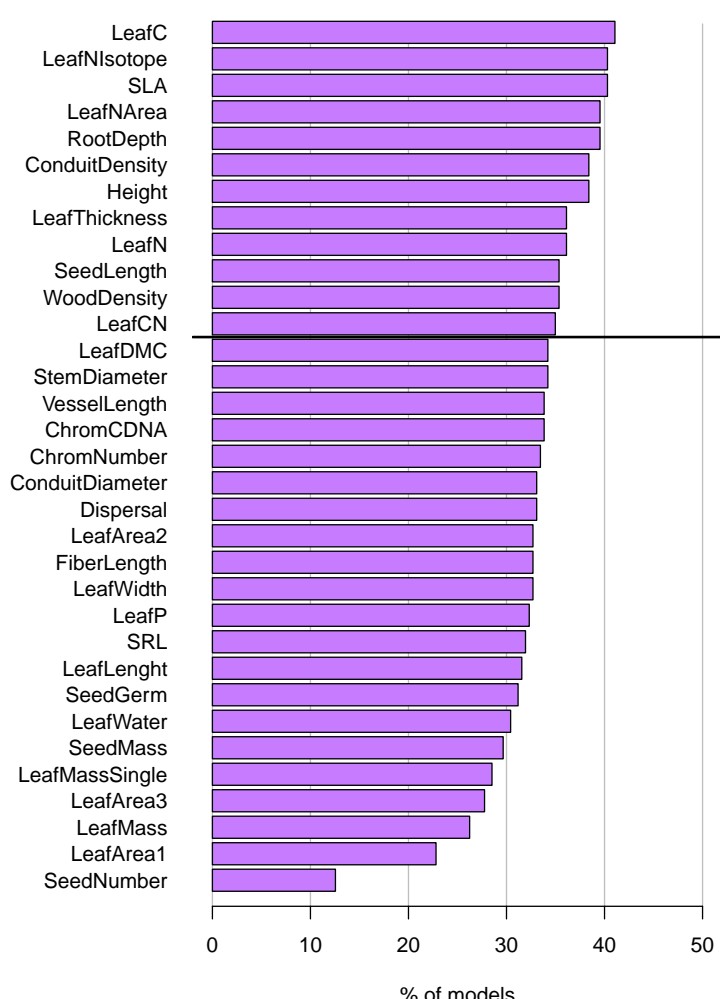

**Figure 3.** Ranking of different traits in cluster analyses. The ranking is based on the percent of models that include the trait in a randomized sensitivity analysis with variable number of randomly selected traits. Analyses were conducted including all traits and all F31 biome maps. The black horizontal line indicates the 12 traits with the highest rank used in trait cluster 1, see sec. 2.4.

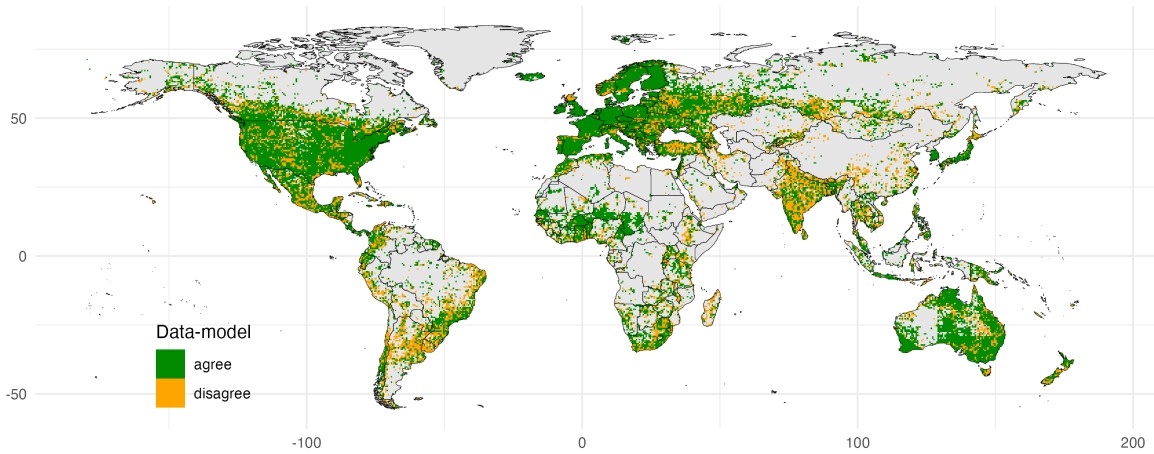

**Figure 4.** Agreement between an observation-based biome map and a map derived from clustering. As an example, the cluster analysis with 12 highly ranked traits of both non-woody and woody plants (see Fig. 3), informed by the Nature Conservancy (2009) biome map (see Fig. S4) was used. The $\kappa$ value for this model was 0.63.

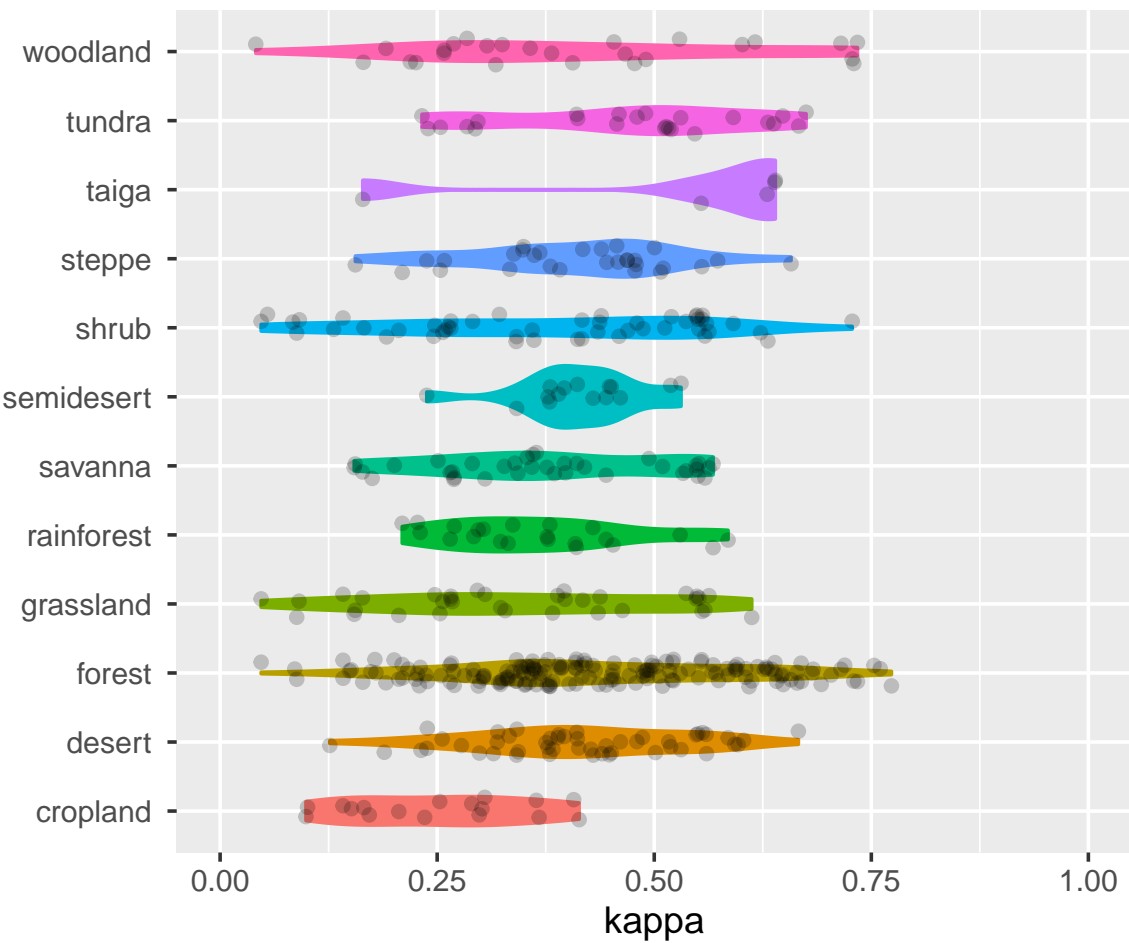

**Figure 5.** Model performance ($\kappa$ values) in different biomes selected by biome type. For this analysis, all biomes that contain the attributes provided in the figure in their names were identified in the F31 biome maps. Then, the $\kappa$ value was calculated for each biome (represented by the points in the figure).

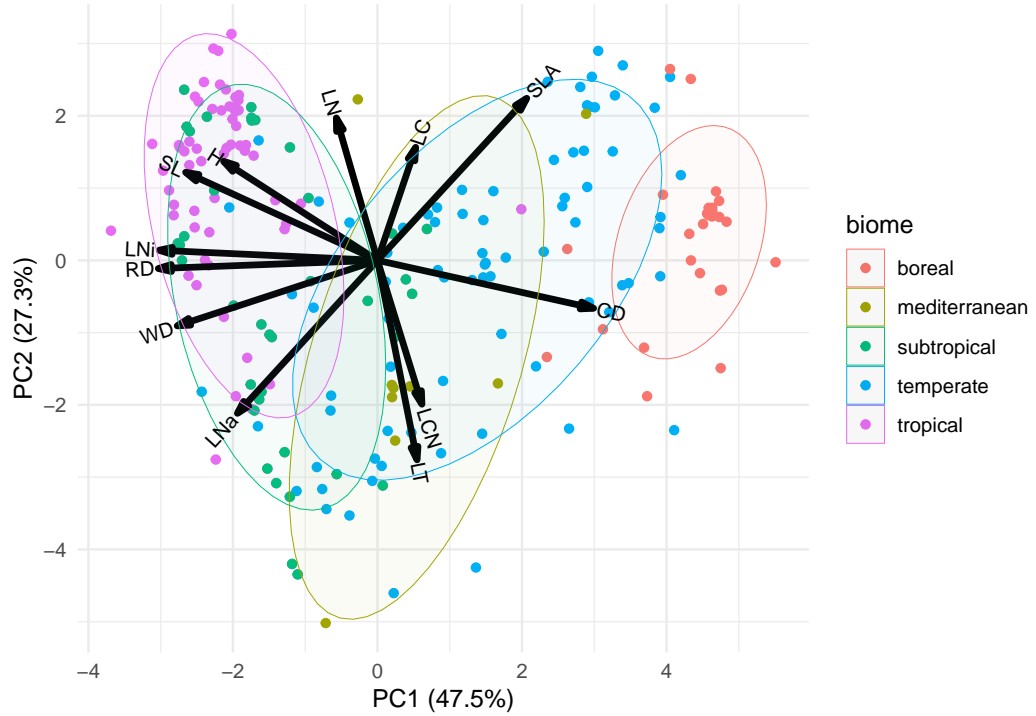

**Figure 6.** Principal component analysis of traits in climatic zones. For the analysis, all biome names in the F31 biome maps (Fischer et al., 2022) containing attributes defining the climatic zone were selected and mean traits were calculated. The PCA was calculated for these trait means. Attributes are represented by different colors. Each point represents one biome type containing the attribute 'boreal', 'mediterranean', 'subtropical', 'temperate' or 'tropical' in one of the F31 biome maps. Results for other biome attributes are provided Figs. S14-S17. Traits are: LC - leaf carbon, LNi - isotopic leaf nitrogen, SLA - specific leaf area, LNa - leaf nitrogen per area, RD - rooting depth, CD - conduit density, H - height, LT - leaf thickness, LN - leaf nitrogen, SL - seed length, WD - wood density, LCN - leaf carbon to nitrogen ratio.

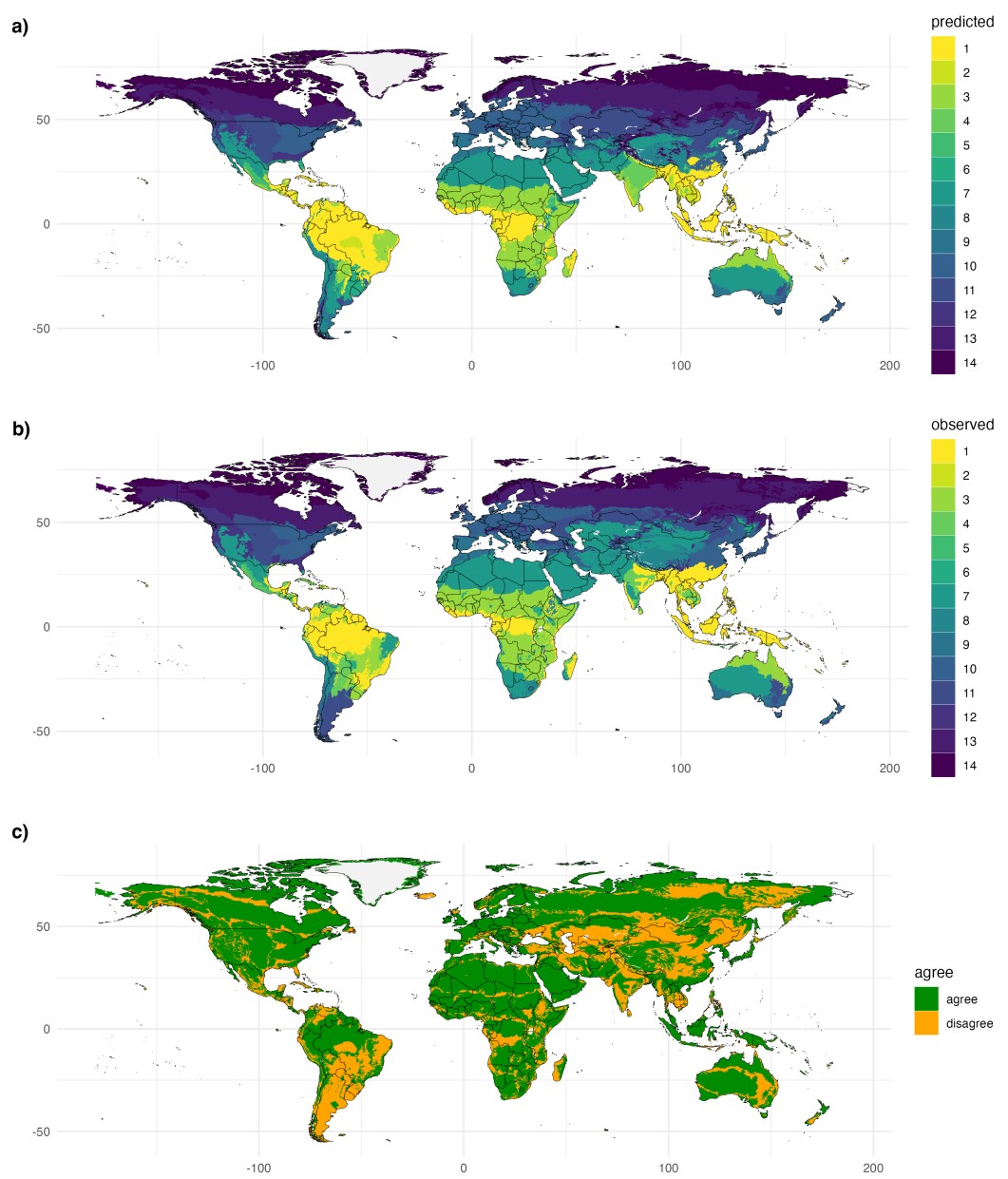

**Figure 7.** Global biomes derived from traits. Using species distribution models and bioclimatic data, biome patterns derived from the spatial coverage of the trait data were extrapolated to the global scale. Here, results from trait cluster 1 and the Nature Conservancy (2009) biome map were used. Biomes: 1, Tropical subtropical moist broadleaf forest; 3, Tropical subtropical grassland savanna and shrub; 4, Tropical subtropical dry broadleaf forest; 5, Tropical subtropical coniferous forest; 6, Flooded grassland and savanna; 7, Desert and xeric shrub; 8, Montane grassland and shrub; 9, Mediterranean forest woodland and scrub; 10, Temperate broadleaf and mixed forest; 11, Temperate grassland savanna and shrub; 12, Temperate conifer forest; 13, Boreal forest/taiga; 14, Tundra.