# Peer review of "Crowd-sourced trait data can be used to delimit global biomes"

_EGUsphere, 2024_

## Author Comment (AC2)

*Scheiter et al. delve into the utilization of crowd-sourced trait data for defining global biomes, investigating the suitability of trait data for biome classification and identifying the most relevant traits for this purpose. While the study is intriguing and significant, I have some comments to offer.*

Thank you for the helpful and constructive comments. As suggested by the reviewer, we will add additional analyses to show the relation between traits and biomes and emphasize the ecological interpretation. We will further move analyses on different PFTs and trait subsets to the supplementary materials such that the main text is more focused. We will also include a schematic figure to illustrate the steps of our analysis. All details are provided in our responses to the comments.

**Major comments:**

- *The findings presented in this study offer valuable insights into the fundamental ecological mechanisms driving plant biogeography and global distribution strategies. However, while the results are intriguing and significant, the interpretation from an ecological perspective could be further emphasized. Additionally, it would be beneficial for the authors to underscore the importance of their study and the implications of their findings, particularly in informing our understanding of global biome classification and its ecological implications. Strengthening the ecological context and emphasizing the practical implications of the findings would significantly enhance the impact and relevance of the study.*

We agree that the ecological interpretation of the results can be strengthened. Currently, the manuscript focuses on the methodological aspects and how the method employed here may be used to generate new ecological findings. We will revise the manuscript with a stronger ecological perspective. Specifically, we will analyze co-variation of traits across biomes in more detail, for example by including plots showing the position of different biomes in the trait space. Therefore, we will conduct a principle component analysis and, based on the loadings of traits and their ranking, illustrate the locations of biomes in the 2-dimensional trait space of selected traits. Such plots will allow us to identify which trait combinations are characteristic for different biomes. We will also revise the discussion to account for these analyses and explain patterns of trait covariation (for example, co-occurrence of tall trees with deep roots and low SLA in tropical forests, indicating intense light competition and the potential for rooting niche separation). We will further underscore the relevance of our results for biome classification by stating more explicitly which traits should be used for classification.

- *The findings regarding the differences in trait distribution and occurrence between biomes are indeed intriguing, but they have been explored somewhat superficially in the manuscript. While I understand that this topic is complex, questions 2 and 4 require a more in-depth exploration throughout the manuscript. As a reader, I was expecting these interesting questions to be thoroughly addressed. For instance, the observed tendency of tropical forests to have plants with greater height could be explored in more detail in the discussion section. One possible explanation for this phenomenon could be light competition.*

Following the suggestion of the reviewer, we will revise the discussion to refer more explicitly to the questions in the introduction. Questions 2 and 4 ask about which and how many traits are required for classification and about characteristic trait values per biome. To deal with question 2, we will strengthen the discussion on traits that turned out to be important in our analysis. These traits agree with previous results on the leaf/plant economic spectrum as well as with previous modeling result that link traits and PFTs (Verheijen et al. 2016). To deal with question 4, we will include additional analysis and figures showing the relation between biomes and traits in the trait space. We will discuss the patterns and explain why different trait combinations co-occur (see also previous point).

- *It would be valuable to delve deeper into why the identified distinguishing traits are significant and what they represent in terms of plant strategies and ecological functions. Hence, you can offer insights into their ecological relevance and their role in shaping biome characteristics. Specifically, you could discuss how these traits contribute to plant adaptation to specific environmental conditions, resource acquisition strategies, and ecosystem functioning. While you have begun to address this topic, emphasizing the ecological significance of the identified traits and highlighting avenues for future research would strengthen the overall argument and underscore the importance of your findings.*

As suggested, we will analyze ecological strategies and co-occurrence of traits in different biomes in more detail in the revised version. Therefore, we will plot the location of biomes in trait space obtained by a principal component analysis. This analysis will illustrate which traits are most relevant to explain the occurrence of different biomes. We will interpret these additional results in the discussion and explain mechanisms causing such patterns (e.g., light competition in dense forests, or water competition, or tradeoffs between acquisitive vs conservative strategies).

- *It's essential to clarify the methodology, especially in section 2.4, to improve understanding for readers.*

We agree that the reasons for conducting this analysis need to be described better. We think that this analysis adds an important aspect to our study, and we will therefor keep it in the revised version of the ms. We will (1) better motivate the analysis, (2) use trait subsets with higher performance (kappa>0.6 in the sensitivity analysis in 2.3, following a comment by reviewer 1), and (3) move Fig. 4 to the appendix and replace it by a novel analysis of the relation between traits and biomes. We will further use this trait set with higher kappa for analyses in 2.5 and 2.6.

- *The conclusion and a significant portion of the discussion focusing on models may seem disconnected from the main purpose of the paper as outlined in the abstract and manuscript title. As a reader, I felt somewhat puzzled. Notably, none of the four questions posed at the end of the introduction mention the theme of modeling. To mitigate this discrepancy, you could consider clarifying from the outset that modeling will be a significant aspect of the study. Alternatively, you may choose to shift the focus of the discussion to emphasize other aspects that were highlighted earlier in the manuscript, aligning more closely with the stated objectives and questions.*

Given that the cluster analyses applied in the study are models, we assume that the comment regarding "none of the four questions … mention … modeling" refers to DGVMs and previous results with aDGVM2 mentioned in the discussion. We agree that this is not the main topic of the manuscript and not reflected in one of the questions. Nonetheless, we found it relevant to compare our results with previous results, both from the analysis of observational data and from modeling.

We will shorten the comparison with aDGVM2 results and make sure that these aspects are not disconnected from the analyses of our study. We will further discuss our results in the context of Verheijen et al. (2016) who used kernel estimation to link traits and PFT and found that similar that similar traits are important. We think that the trait data derived by combining TRY and GBIF are valuable for improving DGVMs, but this is beyond the scope of the presented study and should be addressed in a follow-up study.

*Minor comments:*

*In the abstract the use of the "31 different biome maps" is not clear, is it used to perform the supervised cluster analyses or it is used to evaluate the results from this analyses?*

We will reword the abstract and make clear that the 31 observation-based biome maps are used to inform the supervised cluster analysis of traits into biomes and for the evaluation of the biome maps derived by the cluster analysis. When a single data set is used for model training and testing, it is common to split the data set into a subset for training and a subset for testing. However, as we do not use our models for extrapolations or for predictions with novel data, we argue that our approach is valid.

*Enhancing the fluidity of the introduction by providing more ecological context and emphasizing the significance of generating this type of map would improve the transition to the research questions.*

Many different biome maps are available and were developed using different approaches and data sources. We argue that traits are an additional data source that inform on biophysiological differences among plant communities that can be used generate novel biome maps. This functional trait perspective might provide a more plant-oriented and meaningful understanding and characterization of biomes than, for instance, a biome perspective that is solely based on climate or remotely sensed reflectance. To create such biome maps, we need to conduct two steps, (1) use traits for biome classification and (2) extrapolate from the spatially heterogeneous trait data to a biome map covering the entire land surface by making use of the biomes obtained from clustering, bioclimatic variables and species distribution models. We will better explain this context in the revision.

*l58 and l59 are more appropriate for Materials and Methods section*

We think that the extrapolation to global maps should be kept in the introduction, and we will better motivate this step (see also previous comment). The method section describes how we did this.

*It's not clear in the methods why you "created three different maps for each trait" and why you filtered the observations according to PFTs before spatially aggregating the trait values.*

In line 78, the differentiation of the trait space per PFT might provide added value for biome classification, instead of having only information across all PFT. This step was done before aggregating trait values to be able to conduct the filtering based on the trait data from TRY and keeping as much of the information from TRY as possible.

*In section 2.2 I didn't understand the difference between 3) and 4).*

In case 3, we did not differentiate between woody and non-woody plants to obtain a community trait value per grid cell. In cases 1 and 2, trait values for a grid cell were calculated separately for non-woody and woody plants, resulting in two trait maps. In case 4, we combined the woody and non-woody trait maps in one biome prediction model. Hence, we include more detailed information on PFT-specific trait values.

We will better explain difference between 3 and 4 in the revision.

*In section 2.3, specifying the total number of traits considered would enhance clarity. If I don't miss anything, it is only specified in the abstract.*

By merging TRY and GBIF, data for 33 traits are available and included in the analysis. We will add this information in 2.3.

*It is unclear why the analysis described in section 2.4 was performed if the most important traits were identified in section 2.3.*

The analyses in 2.4 are conducted to make clear that the selection of traits for classification matters and that the selection of the biome map used to inform the cluster analysis matters. This analysis was based on the important traits identified by the sensitivity analysis in 2.3. We will clarify this in the revision and move the results into the supplementary materials.

*Fig. 3: Parallel coordinates plots are already complex visualizations to comprehend, and as it stands, I don't think the figure adds much value, unless it is thoroughly explored and contextualized in the text.*

This figure illustrates that the same set of traits is important (i.e., has a high rank) for all numbers of traits included in the cluster analysis. Similarly, the same set of traits shows low rank. We therefore think that this figure is important as it allows the selection of traits for subsequent analyses (2.4, 2.5, 2.6) using only a subset of traits. We will better explain the figure in the revision and provide context in the text.

*Fig. 5: The background color (gray) on the map is too similar to the light blue, making it difficult to interpret the figure.*

We will change the colors for clarity.

*Why is there such an extensive comparison with the results from aDGVM2? This was not anticipated based on the main questions posed in the Introduction section.*

We will shorten or entirely remove the section on aDGVM2 and therefore focus more on the ecological interpretation of the main results.

*In Section 4.2, you briefly touch on the importance of examining trait covariation, highlighting its significance compared to focusing solely on individual traits. However, given that some of your results seem to directly address this aspect, it would be beneficial to delve deeper into this topic and explicitly connect it with the obtained results.*

We agree that the aspect of trait co-variation was not discussed in depth, and we will strengthen this aspect in the revised version. Our results show co-variation of different traits across biomes in the trait space and a principal component analysis. For example, tropical forests are characterized by the highest vegetation, low SLA, high leaf N and deep roots. These traits represent dominance of tall trees with evergreen phenology (typically associated with low SLA). Tall vegetation also indicated intense light competition. Deep roots indicate that substantial amounts of water percolate into deeper soil layers and may indicate rooting niche separation between tall trees with deep roots and understory vegetation with shallow roots. Deep roots and height can be correlated because tall plants need deep roots for mechanic stability. Shallow roots in boreal forests indicate that those ecosystems are less limited by water or nutrients and more by energy.

*While I acknowledge that analyzing the occurrence and abundance of traits in each biome may not be the primary focus of your manuscript, it presents an opportunity for a valuable complementary analysis. By examining the diversity of traits within each biome, specifically by assessing the variance in their distribution, you can provide additional insights into the functional composition and ecological characteristics of these biomes. This analysis could help elucidate patterns of trait diversity across different environmental gradients and provide a deeper understanding of the ecological processes driving biome differentiation.*

We agree that such an analysis would be very interesting. However, the trait data we used are means for entire 0.5° grid cells such that diversity measures of biomes would represent variability in those mean values and not the full trait space per biome. Therefore, the original TRY data, the merged TRY+GBIF data (i.e., without averaging per grid cell) would be more suitable. In addition to the trait means used in our study, ranges and standard deviations of traits are also available for all traits and could be analyzed to study diversity. Yet, we think that such an analysis would go beyond the scope of the current manuscript, and it should therefore be analyzed in a separate study. We will mention this point in the discussion.

***Writing errors:***

*I believe a parenthesis is missing for the reference in the following sentences:*

1. *"Biomes are commonly used to represent major vegetation formations and to map their biogeographic distributions. Multiple biome maps were developed based on a variety of different data sources Beierkuhnlein and Fischer (2021)."*

2. *"Despite the increasing availability of trait data in databases such TRY (Kattge et al., 2020) and extrapolated global biome maps Wolf et al. (2022); Boonman et al. (2020), a systematic assessment of the performance of traits for biome classification and an identification of the most appropriate traits remain elusive."*

Thanks for spotting those errors, we will correct the brackets.

References

Boonman CCF, Huijbregts MAJ, Benitez-Lopez A, Schipper AM, Thuiller W, Santini L (2022) Trait-based projections of climate change effects on global biome distributions. Diversity and Distributions, 28, 25–37.

Bruelheide H, Dengler J, Purschke O, et al. (2018) Global trait–environment relationships of plant communities. Nature Ecology & Evolution, 2, 1906–1917.

Diaz S, Kattge J, Cornelissen JHC, et al. (2016) The global spectrum of plant form and function. Nature, 529, 167–171.

Verheijen LM, Aerts R, Bönisch G, Kattge J, Van Bodegom PM (2016) Variation in trait trade-offs allows differentiation among predefined plant functional types: implications for predictive ecology. New Phytologist, 209, 563-575.

Wright IJ, Reich PB, Westoby M, et al. (2004) The worldwide leaf economics spectrum. Nature, 428, 821–827.

---

## Author Response (AR1)

**Reviewer 1**

*Scheiter et al present an interesting approach to delineate biomes based on traits information. They use a combination of a number of extensive databases and interesting approaches. Even so, I have a number of major and minor comments.*

We thank the reviewer for the helpful and constructive comments. In response to the comments, we (1) better explained and motivated the analyses on different PFTs and different trait subsets, (2) focused on trait sets with high performance in the main text while moving additional analyses to the supplement, (3) analyzed the relation between traits and biomes in more detail using a PCA to highlight the ecological aspects of our study, and (4) included a schematic figure to illustrate the different steps of our analysis. More details are provided in our responses to the comments.

*Major comments:*

*- A main concern is that some components of the methods do not logically link to each other, which does not help the story line. This specifically applies to two analyses; analyses described in section 2.4 and 2.6.*

*a. The aim of the analysis in 2.4 and how it contributes to the story line remains unclear to me. You create, for each of the four cases (?) a trait set to work with. First of all, I don't understand why that is needed because you already tested the performance of different complete trait sets in 2.3. So, why would you continue working with subsets of traits if you already know how the entire combination of traits works? Now, you seem to throw away most of the information presented in 3.1 instead of building upon them. That does not only seem unnecessary, but also creates biases (because subsets of traits are used; an argument you make yourself in l. 275). Moreover, given the different selection criteria and procedure for different trait sets/cases, interpretation of these results is made even harder (not possible). Also, I don't see the need for doing it. You use it in 2.5 (but then only for the case 4), but I think that also could have been done directly on the best performing trait sets from 2.3. To me, that would have been a much more direct way to test it and with fewer biases.*

As stated by the reviewer, we selected four different sets of traits and conducted cluster analyses informed by each of the 31 biome maps provided by Fischer et al. (2022). In addition, for each trait set, we used traits of different vegetation types (woody, non-woody). We agree that the reasons for conducting this analysis need to be described better, and that we did not use trait combination with the highest performance in the previous sensitivity analysis in 2.3. However, we think that the analysis on trait subsets adds an important aspect to our study for several reasons:

(1) Previous studies used subsets of traits (for example SLA, height, wood density in Boonman et al., 2022). Using the same subsets of traits allows comparisons with previous studies in clearly demonstrates the benefit of including a higher number of traits. Such comparisons highlight the value of merging TRY and GBIF, because the availability of trait information for many traits at large spatial coverage improves the performance of biome classification compared to using only three or four traits.

(2) The sensitivity analyses in 2.3 are based on randomly selected sets of traits, and the analysis shows that some traits have a high rank, i.e., they are represented in many cluster analyses with high kappa (Figs. 2, 3 in original version). Including all traits in the cluster analysis would not allow an identification of relevant traits and a trait ranking. We therefore think that using a systematic selection of traits rather than random selection is reasonable. This selection explicitly builds on the findings of section 3.1 by choosing relevant traits. In addition, we used a PCA to select trait subsets as a more subjective and quantitative method in the original version of the manuscript.

(3) We did not use complete trait sets in 2.3 as stated by the reviewer, but only up to 12 out of 33 traits. Fig. 1 in the original version (Fig. 2 in revision) suggests that kappa saturates as the number of

traits increases such that including more traits does not improve the performance of the cluster analysis substantially. This can be explained by the correlation and redundancy of different traits (Fig. S18 in revision); adding correlated traits does not improve the performance of the clustering substantially and a smaller number of traits is sufficient. This approach also agrees with the common practice of variable selection in statistical modelling. In addition, the clustering algorithm failed when using all 33 traits. Moreover, our results agree with the coordination (redundancy) of traits as found for example in Wright et al. (2004), Diaz et al. (2016) and Bruelheide et al. (2018).

(4) Using a lower number of traits (i.e., 12 of 6 instead of all 33 traits) facilitates the exploration of trait combinations that characterize different biomes (in the analyses in section 2.5) and trait covariation. See also response to comment regarding trait covariation and confusion matrices below.

In the revised version of the manuscript, we kept the analysis on the trait subsets for the above-mentioned arguments and (1) better motivated the analysis based on the points described above, and (2) followed the suggestions made by the reviewer and included trait subsets with higher performance (kappa>0.6 in the sensitivity analysis in 2.3). Therefore, we first selected 12 traits with the highest ranking according to our analysis. From those 12 traits, we then selected 6 traits to reduce complexity. Those 6 traits represent different axes of the plant economic spectrum. Finally, we selected 3 traits (SLA, wood density, height) as in previous studies. We further used the trait set with 12 traits with high rank and a kappa value>0.6 for the analyses in 2.5 and 2.6.

*b: I don't understand why the analysis in 2.6 is useful to do given that you started with using trait maps to fit biome distributions based on complete biome maps. Why would you then derive additional trait maps, using a different procedure with other input data than used elsewhere in the paper. Moreover, you already have 31 biome maps to test performance against. So, why create another one? I would say the study is about understanding biome distributions based on traits (a story already told by the kappa values) and not about predicting/extrapolating biome distributions. In other words, the aim and position of this analysis in the story line is not clear to me.*

We have indeed not been entirely clear on the motivation of this analysis. The observation-based biome maps provided by Fischer et al. (2022) have a spatially continuous coverage, but the trait maps used for the cluster analyses derived by combining TRY and GBIF do have spatial gaps in places where we have no observations (see Fig S1 for spatial coverage of traits, trait data cover ca 50% of the land surface at 0.5° resolution). Therefore, the modeled biome maps derived from the trait map-based cluster analyses do not cover the entire land surface and kappa values only refer to areas with trait data. Thus, we think that a trait-based global biome map with continuous coverage is a valuable outcome of our study, and it represents a novel observation-driven biome map. To create trait-based biome maps with continuous spatial coverage, an additional extrapolation step is required; this is done by using niche models and bioclimatic variables in section 2.6. As we conducted cluster analyses for many different trait combinations supervised by 31 different biome maps, we decided to pick only one combination of a biome map and trait combination with high kappa value for this extrapolation to global scale (Nature Conservancy map and clustering with 12 traits with high rank).

We described and motivated this analysis in more detail in the revision and highlight that an extrapolation is necessary to get from the biome maps from the cluster analysis covering only 50% of the land surface to a full global map. We further added a schematic figure to illustrate the different steps of our analysis. This scheme provides information whether data sets and maps cover the entire land surface or not.

*- Another methodological issue is that I do not understand why the authors thought it important to first use the traits in combination with species distribution models to make trait maps and then to couple those to the biome maps instead of using the locations of the trait observations (g. using the original TRY data, possibly aggregated to the 0.5 degree pixels of the locations) directly? Coupling*

*individual locations can be done to calibrate and validate biome models and avoid the major uncertainties involved in creating/ extrapolating the trait maps, i.e. i don't see that necessity. I wonder to which extent these uncertainties contributed to the low Kappa-values of the predictions.*

We did not use species distribution models to create trait maps (in contrast to other studies that used species distribution models or machine learning methods to extrapolate traits to global scale). Rather trait information from TRY was extrapolated to larger areas by linking observed traits and observed species distributions from GBIF. This method was presented previously (Schiller et al. 2021, Wolf et al., 2022) and showed unprecedented agreement with independent observational data (see section 2.1). The advantage of using these trait maps is that it includes gridded trait data for 33 different traits at much larger spatial scale than the original site data of these traits in TRY (possibly aggregated to 0.5 degree), and all traits are available for the same spatial extent. Further, TRY only represents trait observations obtained from single plants and these observations are known to be not representative of species distributions and plant communities and have a large spatial bias (Kattge et al. 2020). The trait maps obtained from coupling TRY and GBIF represent mean trait values of entire plant communities (Wolf et al. 2022). To be able to do our analyses only with the original TRY site data, instead of gap-filled TRY data, we would need to select sites (or 0.5-degree grid cells) where all 33 traits are available. We suspect that the number of sites would be low and not cover all biomes included in the analysis. This would make it difficult or even impossible to generate global biome maps. We are confident that this analysis can only be conducted because trait data were extrapolated as described in section 2.1.

We better described the reasons for using the trait maps obtained from TRY and GBIF in the revised version of the manuscript and also added this point to the discussion.

*- I had hoped that the authors would have focused more on the ecological interpretation, rather than on the methodological aspects.*

This is a valid point. The original version focused more on methodological aspects. We revised the manuscript according to the suggestions provided below.

*a.: This already starts from the presentation of the data. With slightly different analyses and visualization, potentially a lot more ecological insights might have been gained. For instance, fig 7 presents the mean trait values for some biomes. However, these are individual traits while you tested how different trait combinations allow distinguishing among biomes. However, nowhere we learn about those combinations (their synergies and their redundancies). For instance, I would have liked a so-called confusion matrix to see how for a given trait set, biomes were predicted properly or not and whether this is consistent among the different (best-performing) trait sets: which trait combinations (in addition to single traits) lead to the best predictions? Is that consistent between trait sets? Which trait combinations allow distinguishing e.g. ''subtropical forest'' (just to name one) from other forests? Is that consistent through different trait sets? I think the analysis has the potential to bring such answer and thus ecological understanding, but that remains unexplored.*

As suggested by the referee, we conducted additional analyses to explore trait covariation in different biomes and which trait combinations allow distinguishing biomes. Specifically, we performed principal component analyses to explore where different biomes are located within the trait space. In the PCA, we included traits with high suitability for biome classification according to our ranking. This analysis combined all biome maps provided by the Fischer et al. (2022).

In addition, we provide information on the performance of the clustering per biome and illustrate the results in confusion matrices. A similar confusion matrix was created for the biome map with full global coverage obtained from the species distribution model. Given that we used 31 different biome maps and 33 different traits, it is not possible to provide results for all maps and different trait

subsets. We therefore picked one example with high kappa value: A cluster analysis with the 12 traits with highest rank and the Nature Conservancy biome map (kappa=0.63).

*b: Also the discussion section is now almost entirely focused on the methods instead of the question why the analysis is a useful thing to do and what we learn from it: I would be interested in seeing an ecological interpretation for how and why the results of the analysis help global ecological understanding of biome distribution and functioning. That would have led to a much more interesting story. In the current set-up, the discussion section does not provide much insight.*

We agree that the study has a strong focus on methodological aspects and that there is potential to better highlight ecological aspects. In response to the previous comment, we added analyses on covariation of different traits in biomes and we revised the discussion to interpret these novel results from an ecological perspective. In addition, we gave a better perspective how our methods and results can advance our ecological knowledge and highlight that follow-up studies could focus more on questions regarding trait covariation or functional diversity within biomes.

*Minor comments:*

1. *The abstract does not help to tell the story. The method applied is not clearly explained, i.e. which steps were taken. Also, the role of the 31 biome maps was unclear (in the abstract it seemed an input, while it is used to train the data. Also, a clear take home message is missing: ''we can make biome maps'', but it is not explained why that is important or what is the added value of this study.*

We revised the abstract to address the points raised by the reviewer. The biome maps were used to train the models (we assume that this is meant by "input"). Further, the performance of models was tested against those biome maps. While it is often split into training and testing data, we used the entire data set for training and testing, because we did not use the models/cluster analyses for predictions with novel input data sets. The main aim of the study was indeed to show that traits can be used for biome classification and not so much the ecological implications. Based on the additional analyses described in response to the previous comment, we strengthened the ecological aspects of the abstract and the added value.

2. *I would have appreciated an explanation/argumentation on why choosing these trait maps instead of other maps, i.e. what is the conceptual advantage to these maps? (Also in light of the discussion section where it Is mentioned that different trait maps could have led to different outcomes)*

The advantage of the utilized trait maps is that by combining TRY and GBIF, they cover an unprecedented number and coverage of traits with global cover than covered in the original TRY data (Wolf et al., 2022). Therefore, our analysis is not constrained to a low number of traits commonly used in trait-based analyses (including height, SLA, leaf N, wood density), which allows a more systematic assessment of relevant traits for biome classification and how the performance of classification is related to the number of traits used for clustering. Previous studies where this approach was developed and presented showed better agreement in presenting global patterns of community-weighted traits with independent trait data (sPlotOpen) than other products (Schiller et al. 2021, Wolf et al. 2022). In addition, TRY data are species- and site-specific (Kattge et al. 2020), while the trait data used in our analysis represent entire plant communities. By using GBIF data, we implicitly assume that the observations in GBIF mirror the actual abundances of the species so that we can calculate community weighted means of traits. With the TRY data alone, we cannot assume that observations mirror abundances. We are therefore convinced that using this novel dataset is suitable for the purpose of our study. We added these points to the discussion.

*3. With some self-advertisement; Verheijen et al. NewPhytol 209: 563-575 also evaluated which traits could be used best to distinguish among biomes, albeit which is much smaller dataset and fewer traits (And a different method). Interestingly, similar traits popped up as important.*

Thanks for sharing this resource, which is indeed very valuable to back-up our findings. We cited the paper in the introduction and discussion as suggested and refer to the results presented in this study. It is indeed interesting, that similar traits are important in both studies.

*4. At some parts in the methods section it is unclear whether all individual 31 biome maps were tested independently or whether an aggregate was used. It seems that you tested each map individually, but a better explanation throughout the methods on this would have been appreciated.*

All maps were tested independently, and no aggregation was used. As the biome maps were created by different methods and using different data sources, aggregation is challenging. We revised the text to make this clear.

*5. The role and use of PFTs is confusing to me. You derive different trait maps depending on growth form/PFT and clustered them into four cases in 2.2. Then, it seems that trait clusters were made for each of those four cases (if I understand the methods correctly) in 2.3. Then, later in 2.3 you seem to continue with the outcomes of case4 only. This is not clearly described and also the reason why you do this, remains unclear to me. Also the role and use of PFTs is confusing to me. In general, i don't think you tested different combinations of PFTs, but trait maps of woody vs non-woody vegetation.*

Yes, the analysis in 2.3 was conducted as described by the reviewer, and we conducted cluster analyses for four different cases. In further analyses, we continued with case 4 only because our analyses showed that kappa values are higher when traits of both non-woody and woody plants are used. For further analyses we aimed at using traits and trait combinations with high kappa values. The trait values of woody and non-woody plants were derived by combining only those TRY and GBIF observations that correspond to specific PFTs according to the TRY database and represent mean trait values of all non-woody and/or woody species within the 0.5° grid cells. We described this point more clearly in the revised version.

Regarding the last point of the comment: for the first two cases with only non-woody or only woody plants, we test indeed non-woody vs woody vegetation. We tested if the performance of cluster analyses using only non-woody plants (or woody plants) are better in grass-dominated biomes (or forests, respectively), but there was no robust pattern (results not shown in manuscript). For the two other cases, both non-woody and woody plants are included but they were aggregated in a single trait value representing all non-woody and woody species in a grid cell (in case 3) or considered separately (in case 4). Using these two approaches, we could show that separately considering the trait space of non-woody and woody plants enhances the differentiation of biomes.

We better explained the reasons for using different PFTs in the revised version. We moved the results from this analysis into the supplement (particularly Fig. 4 of original version) to focus on one case (case 4) in the main text.

*6. I understand you have a personal interest to compare to aDGVM2 results, but to me those comparisons do not add much to the story.*

In the discussion, we compare our results to previous results, derived from both empirical analyses and modeling. We therefore kept the aDGVM2 modeling in the discussion but shortened it substantially. We also refer to the Verheijen et al. (2016) study in the revised version.

*7. The title of section 4.2 is strange and seems wrong.*

This section describes how the choice of the observation-based biome map used to train the supervised biome classification (here 31 different maps provided by Fischer et al., 2022) influences the performance of the cluster analysis and agreement of the observation-based biome map. Therefore, we think that the title is correct. Nonetheless, we revised it for clarity: "Selection of biome map for clustering influences model performance".

*8. I am surprised that the kappa does not go beyond 0.6. I had hoped that higher kappas would be feasible. This is not discussed, but I would be interested (e.g. instead of the current 4.3) if such aspects of model performance would be discussed and interpreted.*

Good point. The kappa values below 0.6 in Fig. 4 of the original version can be explained by (1) the selection of a subset of traits and (2) the aggregation of all biomes. When using different trait combinations and calculating kappa per biome, higher kappa values can be obtained. For example, the kappa goes beyond 0.6 in the sensitivity analysis in when at least 8 non-woody and woody plant traits (i.e., 16 variables in total) are included in the analysis. When considering biomes individually (Figs. 6, S3-S6 in original version, Figs. 5, S5-S8 in revision), kappa values can exceed 0.75 (for example tropical forest or evergreen forest). In response to a previous comment, we included an analysis with a trait combination with a higher kappa value. We used the 12 traits with the highest suitability according to our ranking and the selected the Nature Conservancy map for case studies. For this trait-biome map combination, the kappa value was 0.63, ranging between 0.25 and 0.79 for different biomes (Table 1 in revision). As suggested, we discuss why the kappa values do not exceed 0.6 considerably. We attribute this to different coverage of trait data and species distribution data in different biomes and to differences in the accuracy of combining TRY and GBIF data.

*9. With respect to figure 3; what does it tell us if ranks of certain traits vary with number of traits (while ranks of other don't)? i.e. what is the message/understanding this figure gives?*

The figures shows that the trait ranking is robust for varying numbers of traits included in the cluster analysis, i.e., the same sets of traits show high or low rank for different numbers. This implies that for biome classification it is more important to select an appropriate set of traits instead of selecting a high number of (randomly selected) traits. Selecting a high number of inappropriate traits can imply low performance of the clustering. Traits are often coordinated and/or correlated and therefore different trait sets may lead to different ranks and performance. The ranking allows systematic selection of appropriate traits for biome classification, and the selection of traits in the analyses described in section 2.4 is based on this ranking. We clarified the value of the figure and the selection of traits in 2.4 in the revised version of the manuscript. As suggested by referee 2, we moved the figure to the supplement.

References

Boonman CCF, Huijbregts MAJ, Benitez-Lopez A, Schipper AM, Thuiller W, Santini L (2022) Trait-based projections of climate change effects on global biome distributions. Diversity and Distributions, 28, 25–37.

Bruelheide H, Dengler J, Purschke O, et al. (2018) Global trait–environment relationships of plant communities. Nature Ecology & Evolution, 2, 1906–1917.

Diaz S, Kattge J, Cornelissen JHC, et al. (2016) The global spectrum of plant form and function. Nature, 529, 167–171.

Fischer JC, Walentowitz A, Beierkuhnlein C (2022) The biome inventory - standardizing global biogeographical land units. Global Ecology and Biogeography, 31, 2172–2183.

Kattge J, Bönisch G, Diaz S, et al. (2020) TRY plant trait database - enhanced coverage and open access. Global Change Biology, 26, 119–188.

Schiller C, Schmidtlein S, Boonman C, Moreno-Mariinez A, Kattenborn T (2021) Deep learning and citizen science enable automated plant trait predictions from photographs, Scientific Reports, 11, 16395

Verheijen LM, Aerts R, Bönisch G, Kattge J, Van Bodegom PM (2016) Variation in trait trade-offs allows differentiation among predefined plant functional types: implications for predictive ecology. New Phytologist, 209, 563-575.

Wolf S, Mahecha MD, Sabatini FM, et al. (2022) Citizen science plant observations encode global trait patterns. Nature Ecology & Evolution.

Wright IJ, Reich PB, Westoby M, et al. (2004) The worldwide leaf economics spectrum. Nature, 428, 821–827.

**Reviewer 2:**

*Scheiter et al. delve into the utilization of crowd-sourced trait data for defining global biomes, investigating the suitability of trait data for biome classification and identifying the most relevant traits for this purpose. While the study is intriguing and significant, I have some comments to offer.*

Thank you for the helpful and constructive comments. As suggested by the reviewer, we added additional analyses (PCAs) to show the relation between traits and biomes and emphasized the ecological interpretation. We further moved analyses on different PFTs and trait subsets to the supplementary materials such that the main text is more focused. We included a schematic figure to illustrate the steps of our analysis. All details are provided in our responses to the comments.

*Major comments:*

- *The findings presented in this study offer valuable insights into the fundamental ecological mechanisms driving plant biogeography and global distribution strategies. However, while the results are intriguing and significant, the interpretation from an ecological perspective could be further emphasized. Additionally, it would be beneficial for the authors to underscore the importance of their study and the implications of their findings, particularly in informing our understanding of global biome classification and its ecological implications. Strengthening the ecological context and emphasizing the practical implications of the findings would significantly enhance the impact and relevance of the study.*

We agree that the ecological interpretation of the results can be strengthened. The original version of the manuscript focused on the methodological aspects and how the method employed here may be used to generate new ecological findings. We revised the manuscript with a stronger ecological perspective. Specifically, we analyzed co-variation of traits across biomes in more detail, by conducting a principal component analysis. We added biplots of those PCA to the manuscript to show the location of biomes in trait space. Such plots allow us to identify which trait combinations

are characteristic for different biomes. We also revised the discussion to account for these analyses and explain patterns of trait covariation (for example, co-occurrence of tall trees with deep roots and low SLA in tropical forests, indicating intense light competition and the potential for rooting niche separation). We further underscored the relevance of our results for biome classification by stating more explicitly which traits should be used for classification.

- *The findings regarding the differences in trait distribution and occurrence between biomes are indeed intriguing, but they have been explored somewhat superficially in the manuscript. While I understand that this topic is complex, questions 2 and 4 require a more in-depth exploration throughout the manuscript. As a reader, I was expecting these interesting questions to be thoroughly addressed. For instance, the observed tendency of tropical forests to have plants with greater height could be explored in more detail in the discussion section. One possible explanation for this phenomenon could be light competition.*

Following the suggestion of the reviewer, we revised the discussion to refer more explicitly to the questions in the introduction. Questions 2 and 4 ask about which and how many traits are required for classification and about characteristic trait values per biome. To deal with question 2, we strengthened the discussion on traits that turned out to be important in our analysis. These traits agree with previous results on the leaf/plant economic spectrum as well as with previous modeling result that link traits and PFTs (Verheijen et al. 2016). To deal with question 4, we included additional analyses and figures showing the relation between biomes and traits in the trait space. We discussed the patterns and explained why different trait combinations co-occur (see also previous point).

- *It would be valuable to delve deeper into why the identified distinguishing traits are significant and what they represent in terms of plant strategies and ecological functions. Hence, you can offer insights into their ecological relevance and their role in shaping biome characteristics. Specifically, you could discuss how these traits contribute to plant adaptation to specific environmental conditions, resource acquisition strategies, and ecosystem functioning. While you have begun to address this topic, emphasizing the ecological significance of the identified traits and highlighting avenues for future research would strengthen the overall argument and underscore the importance of your findings.*

As suggested, we analyzed ecological strategies and co-occurrence of traits in different biomes in more detail in the revised version. Therefore, we conducted principal component analyses and created biplots to illustrate the location of biomes in trait space. This analysis illustrates which traits are most relevant to explain the occurrence of different biomes. We interpret these additional results in the discussion and explain mechanisms causing such patterns (e.g., light competition in dense forests, or water competition, or tradeoffs between acquisitive vs conservative strategies).

- *It's essential to clarify the methodology, especially in section 2.4, to improve understanding for readers.*

We agree that the reasons for conducting this analysis need to be described better. We think that this analysis adds an important aspect to our study, and we therefore kept it in the revised version of the manuscript. In the revision, we (1) better motivated the analysis, (2) used trait subsets with higher performance (12 traits with highest ranking and kappa>0.6 in the sensitivity analysis in 2.3), and (3) moved Fig. 4 to the appendix and replace it by a novel analysis of the relation between traits and biomes. We further used this trait set with higher kappa for analyses in 2.5 and 2.6.

- *The conclusion and a significant portion of the discussion focusing on models may seem disconnected from the main purpose of the paper as outlined in the abstract and manuscript title.*

*As a reader, I felt somewhat puzzled. Notably, none of the four questions posed at the end of the introduction mention the theme of modeling. To mitigate this discrepancy, you could consider clarifying from the outset that modeling will be a significant aspect of the study. Alternatively, you may choose to shift the focus of the discussion to emphasize other aspects that were highlighted earlier in the manuscript, aligning more closely with the stated objectives and questions.*

Given that the cluster analyses applied in the study are models, we assume that the comment regarding "none of the four questions … mention … modeling" refers to DGVMs and previous results with aDGVM2 mentioned in the discussion. We agree that this is not the main topic of the manuscript and not reflected in one of the questions. Nonetheless, we found it relevant to compare our results with previous results, both from the analysis of observational data and from modeling.

We shortened the comparison with aDGVM2 results and made sure that these aspects are not disconnected from the analyses of our study. We further discussed our results in the context of Verheijen et al. (2016) who used kernel estimation to link traits and PFT and found that similar that similar traits are important. We think that the trait data derived by combining TRY and GBIF are valuable for improving DGVMs, but this is beyond the scope of the presented study and should be addressed in a follow-up study.

**Minor comments:**

*In the abstract the use of the "31 different biome maps" is not clear, is it used to perform the supervised cluster analyses or it is used to evaluate the results from these analyses?*

We revised the abstract and made clear that the 31 observation-based biome maps are used to inform the supervised cluster analysis of traits into biomes and for the evaluation of the biome maps derived by the cluster analysis. When a single data set is used for model training and testing, it is common to split the data set into a subset for training and a subset for testing. However, as we do not use our models for extrapolations with novel data, we argue that our approach is valid.

*Enhancing the fluidity of the introduction by providing more ecological context and emphasizing the significance of generating this type of map would improve the transition to the research questions.*

Many different biome maps are available and were developed using different approaches and data sources. We argue that traits are an additional data source that inform on biophysiological differences among plant communities that can be used generate novel biome maps. This functional trait perspective might provide a more plant-oriented and meaningful understanding and characterization of biomes than, for instance, a biome perspective that is solely based on climate or remotely sensed reflectance. To create such biome maps, we need to conduct two steps, (1) use traits for biome classification and (2) extrapolate from the spatially heterogeneous trait data to a biome map covering the entire land surface by making use of the biomes obtained from clustering, bioclimatic variables and species distribution models. We better explained this context in the revision.

*l58 and l59 are more appropriate for Materials and Methods section*

We think that the extrapolation to global maps should be kept in the introduction. We better motivated this step (see also previous comment). We revised the method section to better motivate and describe this modeling step.

*It's not clear in the methods why you "created three different maps for each trait" and why you filtered the observations according to PFTs before spatially aggregating the trait values.*

In line 78 of the original version, the differentiation of the trait space per PFT (non-woody vs woody species) might provide added value for biome classification, instead of having only information across all species. This step was done before aggregating trait values to be able to conduct the filtering based on the trait data from TRY and keeping as much of the information from TRY as possible.

*In section 2.2 I didn't understand the difference between 3) and 4).*

In case 3, we did not differentiate between woody and non-woody plants to obtain a community trait value per grid cell but use trait information of all species. In cases 1 and 2, trait values for a grid cell were calculated separately for non-woody and woody plant species, resulting in two trait maps per trait. In case 4, we combined the woody and non-woody trait maps (cases 1 and 2) in one biome prediction model. Hence, we include more detailed information on PFT-specific trait values. We better explained the different cases in the revision.

*In section 2.3, specifying the total number of traits considered would enhance clarity. If I don't miss anything, it is only specified in the abstract.*

By merging TRY and GBIF, data for 33 traits are available and included in the analysis. We added this information in 2.3.

*It is unclear why the analysis described in section 2.4 was performed if the most important traits were identified in section 2.3.*

The analyses in 2.4 are conducted to make clear that the selection of traits for classification matters and that the selection of the biome map used to inform the cluster analysis matters. This analysis was based on the important traits identified by the sensitivity analysis in 2.3. We modified the selection of traits and used the 12 traits with the highest rank in the sensitivity analysis. Selecting these traits ensures high kappa values for subsequent analyses in 2.5 and 2.6. Then we reduced the number of traits to 6 and 3 traits, to remove redundancy while keeping traits representing different axes of the plant economic spectrum. We better motivated the selection of traits, re-did the analyses in 2.4, 2.5 and 2.6, and revised the figures.

*Fig. 3: Parallel coordinates plots are already complex visualizations to comprehend, and as it stands, I don't think the figure adds much value, unless it is thoroughly explored and contextualized in the text.*

This figure illustrates that the same set of traits is important (i.e., has a high rank) for all numbers of traits included in the cluster analysis. Similarly, the same set of traits shows low rank. We therefore think that this figure is important as it allows the selection of traits for subsequent analyses (2.4, 2.5, 2.6) using only a subset of traits. We better explained the figure in the revision and provided more context in the text. The figure was moved into the supplement.

*Fig. 5: The background color (gray) on the map is too similar to the light blue, making it difficult to interpret the figure.*

We changed the colors for clarity.

*Why is there such an extensive comparison with the results from aDGVM2? This was not anticipated based on the main questions posed in the Introduction section.*

We shortened the section on aDGVM2 and therefore focus more on the ecological interpretation of the main results.

*In Section 4.2, you briefly touch on the importance of examining trait covariation, highlighting its significance compared to focusing solely on individual traits. However, given that some of your results seem to directly address this aspect, it would be beneficial to delve deeper into this topic and explicitly connect it with the obtained results.*

We agree that the aspect of trait co-variation was not discussed in depth, and we strengthened this aspect in the revised version. We conducted a principal component analysis to study the location of different biomes in trait space. Our results show co-variation of different traits across biomes in the trait space. For example, tropical forests are characterized by the highest vegetation, low SLA, high leaf N and deep roots. These traits represent dominance of tall trees with evergreen phenology (typically associated with low SLA). Tall vegetation also indicated intense light competition. Deep roots indicate that substantial amounts of water percolate into deeper soil layers and may indicate rooting niche separation between tall trees with deep roots and understory vegetation with shallow roots. Deep roots and height can be correlated because tall plants need deep roots for mechanic stability. Shallow roots in boreal forests indicate that those ecosystems are less limited by water or nutrients and more by energy. We added these points to the discussion.

*While I acknowledge that analyzing the occurrence and abundance of traits in each biome may not be the primary focus of your manuscript, it presents an opportunity for a valuable complementary analysis. By examining the diversity of traits within each biome, specifically by assessing the variance in their distribution, you can provide additional insights into the functional composition and ecological characteristics of these biomes. This analysis could help elucidate patterns of trait diversity and provide a deeper understanding of the ecological processes driving biome differentiation.*

We agree that such an analysis would be very interesting. However, the trait data we used are means for entire 0.5° grid cells such that diversity measures of biomes would represent variability in those mean values and not the full trait space per biome. Therefore, the original TRY data, the merged TRY+GBIF data (i.e., without averaging per grid cell) would be more suitable. In addition to the trait means used in our study, ranges and standard deviations of traits are also available for all traits and could be analyzed to study diversity. Yet, we think that such an analysis would go beyond the scope of the current manuscript, and it should therefore be analyzed in a separate study. We mentioned this point in the conclusions.

*Writing errors:*

*I believe a parenthesis is missing for the reference in the following sentences:*

1. *"Biomes are commonly used to represent major vegetation formations and to map their biogeographic distributions. Multiple biome maps were developed based on a variety of different data sources Beierkuhnlein and Fischer (2021)."*

2. *"Despite the increasing availability of trait data in databases such TRY (Kattge et al., 2020) and extrapolated global biome maps Wolf et al. (2022); Boonman et al. (2020), a systematic assessment of the performance of traits for biome classification and an identification of the most appropriate traits remain elusive."*

Thanks for spotting those errors, we corrected the brackets.

References

Boonman CCF, Huijbregts MAJ, Benitez-Lopez A, Schipper AM, Thuiller W, Santini L (2022) Trait-based projections of climate change effects on global biome distributions. Diversity and Distributions, 28, 25–37.

Bruelheide H, Dengler J, Purschke O, et al. (2018) Global trait–environment relationships of plant communities. Nature Ecology & Evolution, 2, 1906–1917.

Diaz S, Kattge J, Cornelissen JHC, et al. (2016) The global spectrum of plant form and function. Nature, 529, 167–171.

Verheijen LM, Aerts R, Bönisch G, Kattge J, Van Bodegom PM (2016) Variation in trait trade-offs allows differentiation among predefined plant functional types: implications for predictive ecology. New Phytologist, 209, 563-575.

Wright IJ, Reich PB, Westoby M, et al. (2004) The worldwide leaf economics spectrum. Nature, 428, 821–827.

---

## Referee Report (RR1)

**Referee report**

Naixin Fan[1,2]

[1]TUD Dresden University of Technology, Faculty of Environmental Sciences, Junior Professorship in Environmental Remote Sensing, Helmholtzstr. 10, 01069.
[2]Max Planck Institute for Biogeochemistry, Hans Knöll Strasse 10, 07745 Jena, Germany

**Recommendation**

I recommend accepting the manuscript for two reasons: (1) it provides significant insights into the key traits that statistically distinguish biomes, and (2) the authors, Scheiter et al., have carefully addressed the reviewers' concerns, leading to substantial improvements in the revised version.

**Suggestions for revision**

1. The introduction can be better structured. The ecological importance of the concept of biomes is not properly introduced, if not neglected. Adding a few sentences at the beginning would greatly help clarify the motivation for your study and draw readers' attention. Similarly, another key concept used throughout the paper, 'trait,' is not introduced. Audiences other than scientists who work in ecology might have difficulty understanding this term at first glance. Additionally, the connection between the previous works cited in the introduction is not very clear. In my opinion, there is still much room for improvement in the introduction.

2. A similar issue exists in the 'Data' section. It would be clearer if the author first introduced what is contained in the TRY database, for example.

3. The selection of subset traits in Section 2.4 can be better explained. Why is a certain number of subsets selected from the 33 candidates? I recommend incorporating your responses to Reviewer #1 into the manuscript to better explain the connection.

---

## Author Response (AR2)

Dear editor, dear reviewers,

thank you for the positive assessment and the opportunity to submit a revised version of the manuscript. Following the comments and suggestions of the reviewers, we revised the manuscript, particularly by better explaining traits and the biome concept, and by revising selected sections of the methods.

We are looking forward to your decision.

Yours sincerely,

Simon Scheiter and co-authors

*Editor*

*Dear authors:*
*Thank you for your contribution to Biogeosciences.*
*Your revised manuscript has been evaluated by two anonymous referees. They judged that it is of good scientific significance and quality and has been improved from the previous manuscript. They recommended that minor amendments be made before it is accepted for publication. Please carefully consider the recommendations of the referees and provide your responses to them.*

Thank you for the positive feedback. We considered all reviewer recommendations in our revision. Our responses are provided below.

*Review 1*

*I appreciate the detailed responses given by the authors on my previous comments. These responses in combination with the changes made in the manuscript convincingly deal with most of the comments made. Two points remain to my point of view, but I leave it to the authors to decide whether and how they would like to resolve these points.*

Thanks for this positive feedback.

*I am still not convinced by the added value of the map created. As I indicated in my original review, and that did not change by the reply provided: I don't see the added values of this analysis and product given the aims of the paper and to me it does not seem to fit the story line, while it adds to the overall complexity of the paper (see my second point). Moreover, 50% of the map is based on extrapolation. While that is common to many other maps too, other maps always include a validation step to evaluate the extrapolation. No such validation was included here. There is thus no way to test the validity of the map.*

We agree that the global map adds to the complexity of the manuscript. Nonetheless, we think that it is an important outcome of our study, as it shows that we cannot only reproduce biomes for areas where trait data is available, but that we can also extrapolate to the global scale. We therefore prefer to keep the analysis in the manuscript.

Regarding the validation, we compared the extrapolated biome map to the observation-based biome map used to create the SDM using kappa statistics and TSS as well as by creating a confusing matrix (Table 2). We reworded the methods in section 2.6 and the results in section 3.4 to make this clearer.

*Also, while figure 1 helps to better maintain the overview of the design and steps taken in the paper, it is still quite complicated and a lot of concentration of the reader is needed to keep track of all terminology and steps. This cannot be solved easily, but is a point of attention.*

We checked the figure. Yet, as we decided to keep the global extrapolation in the manuscript (see previous points) we did not find ways for simplification.

*Review 2*

*1. The introduction can be better structured. The ecological importance of the concept of biomes is not properly introduced, if not neglected. Adding a few sentences at the beginning would greatly help clarify the motivation for your study and draw readers' attention. Similarly, another key concept used throughout the paper, 'trait,' is not introduced. Audiences other than scientists who work in ecology might have difficulty understanding this term at first glance. Additionally, the connection between the previous works cited in the introduction is not very clear. In my opinion, there is still much room for improvement in the introduction.*

Following the suggestions of the reviewer, we added (1) statements to highlight the importance of biomes and the biome concept, and (2) a definition of traits in the introduction. We also checked the cited literature and strengthened the connection.

*2. A similar issue exists in the 'Data' section. It would be clearer if the author first introduced what is contained in the TRY database, for example.*

We revised the 'Data' section 2.1 and added information on the TRY and GBIF databases.

*3. The selection of subset traits in Section 2.4 can be better explained. Why is a certain number of subsets selected from the 33 candidates? I recommend incorporating your responses to Reviewer #1 into the manuscript to better explain the connection.*

As suggested by the reviewer, we revised section 2.4 to better explain why we selected subsets from the 33 available traits. This revision was based on the points from our previous responses to reviewer #1.